

# Sharp increase of Saharan dust intrusions over the Western Mediterranean and Euro-Atlantic region in winters 2020-2022 and associated atmospheric circulation

Emilio Cuevas-Agulló[1], David Barriopedro[2], Rosa Delia García[1,3], Silvia Alonso-Pérez[4,5], Juan Jesús González-Alemán[6], Ernest Werner[7], David Suárez[8], Juan José Bustos[1], Gerardo García-Castrillo[7], Omaira García[1], África Barreto[1], Sara Basart[9,10]

[1]Izaña Atmospheric Research Center (IARC), State Meteorological Agency (AEMET), Santa Cruz de Tenerife, 38001 Spain
[2]Instituto de Geociencias (IGEO), Consejo Superior de Investigaciones Científicas - Universidad Complutense de Madrid (CSIC-UCM), Madrid, 28040, Spain
[3]TRAGSATEC, Madrid, Spain
[4]Departamento de Ingeniería Industrial, Universidad de La Laguna (ULL), La Laguna (Tenerife), 38002, Spain
[5]Instituto Universitario de Estudios de las Mujeres, Universidad de La Laguna (ULL), La Laguna (Tenerife), 38002, Spain
[6]Modeling Area-Department of Development and Applications, State Meteorological Agency (AEMET), Madrid, 28040, Spain
[7]Territorial Delegation of AEMET in Catalonia, State Meteorological Agency (AEMET), Barcelona, 08071, Spain
[8]Territorial Delegation of AEMET in The Canary Islands, State Meteorological Agency of Spain (AEMET), Las Palmas, 35017, Spain
[9]Barcelona Supercomputing Center (BSC), Barcelona, 08034, Spain
[10]Now at the World Meteorological Organization (WMO), Geneva, 1201, Switzerland

*Correspondence to*: Sara Basart (sbasart@wmo.int)

**Abstract**

During the winters of the 2020-2022 period, several intense North African dust intrusions affected Europe. Some of them displayed a duration never recorded before. They were referred to as exceptional by several international operational and research institutions considering that wintertime is the season with minimum dust activity in the Mediterranean and Europe. These anomalous winter events with origin in North Africa largely affected western Mediterranean. The main objective of the present work is to analyse the atmospheric drivers (synoptic and large-scale environments) of wintertime (from January to March) dust events over the region covering North Africa, the Western Mediterranean and the Euro-Atlantic during the period 2003-2022. Overall, our results indicate large interannual variability over the study period. A dust catalogue of dust events identified by aerosols retrievals from satellite and aerosol reanalysis products shows a very irregular record and large differences between winter months. The analyses demonstrate a positive anomaly in dust concentration and maximum altitude during the dust events of 2020-2022 in comparison with those of previous years (2003-2019). Winter dust events over western Mediterranean are associated with enhanced blocking activity over the Euro-Atlantic sector, which favours the obstruction of



the westerlies and the occurrence of cut-off lows at subtropical latitudes. However, these high-pressure systems can exhibit a large variety of configurations, including meridional dipole blocking patterns with poleward shifted jets or Mediterranean subtropical ridges with an intensified mid-latitude jet. The former was more frequent during the reference 2003-2019 period, whereas the latter was relatively common during the anomalous 2020-2022 period.

## 1 Introduction

Mineral dust, when considering its mass, is the most abundant type of atmospheric aerosol found over the ocean and continents, representing about 75% of the global aerosols present in the atmosphere (Kinne et al., 2006; Wu et al., 2020). Dust affects the climate system by interacting with longwave and shortwave radiation, as well as contributing to the formation of cloud condensation nuclei (CCN) and ice nucleating particles (INP) (e.g. Szopa et al., 2021). Recent studies revealed that the magnitude of radiative forcing due to mineral dust is small (Kok et al., 2017; Ryder et al., 2018) because dust particles seem to be larger than previously thought (Kok et al., 2017; Szopa et al., 2021). This new finding, along with the fact that global models underestimate the amount of coarse dust in the atmosphere (Adebiyi and Kok, 2020) raises the possibility that dust emissions may even be warming the climate system (Szopa et al., 2021). Primary emissions of mineral dust from deserts are the most important contributor to the aerosol budget in the Middle East, Eurasia, and Africa, accounting for 40–70 % of the annual average atmospheric concentrations (Szopa et al., 2021). In large regions of the Earth, mineral dust exerts a great impact on ecosystems (UNEP, 2021), human health (Aghababaeian et al., 2021; Kinni et al., 2021) and other socio-economic sectors such as energy and transportation (Cuevas et al., 2021; ESCAP, 2021; Monteiro et al 2022).

The Sahara and surrounding regions contribute to more than half of global dust emissions (Kok et al., 2021). Dust mobilization and transport in northern Africa shows high variability at different time scales: diurnal (Knippertz, 2008; Cuesta et al., 2009; Fiedler et al., 2014), intraseasonal (Ashpole and Washington, 2013; Cuevas et al., 2017), seasonal (Barnaba and Gobbi, 2004; Israelevich et al., 2012), interannual (Rodríguez et al., 2014; Tegen et al., 2013; Wagner et al., 2016) and multidecadal (Wang et al., 2015) time scales. Thunderstorms and cyclones can produce high-speed winds that lift the dust and transport it thousands of miles away through the air. African dust can travel around the globe to parts of Europe, South America, Central America, the Caribbean, and the United States (e.g., Shepherd et al., 2016). Mineral dust mobilized in the Sahara and North Africa often reaches the Mediterranean basin and Canary Islands (Moulin et al., 1998; Basart et al., 2009, Varga et al., 2014, Gkikas et al., 2015; Cuevas et al., 2021), affecting air quality and causing exceedances of the thresholds established by the World Health Organisation (WHO) Air Quality Guidelines (WHO, 2021). Particularly in southern European countries, African dust intrusions causes the exceedance of the European Union Air Quality Directive (Kuula et al., 2021) for particulate matter (PM) (Gerasopoulos et al., 2006; Escudero et al., 2007; Querol et al., 2009; Basart et al., 2012; Pey et al., 2013; Marconi et al., 2014). Also, extreme dust events in Europe have had enormous impacts on socio-economic activities such as energy and transportation with consequently economic losses (e.g., Monteiro et al 2022). Otherwise, Saharan dust deposition fertilizes the



Mediterranean Sea triggering marine phytoplankton growth, but also harmful cyanobacterial blooms (e.g., Guerzoni et al., 1999; Gallisai et al., 2014; Lagaria et al., 2017).

The challenge of mitigating the impacts of sand and dust storms (SDS) is recognised globally. The United Nations (UN) agencies are promoting measures to confront the problem and their inclusion in national policies through the UN Coalition for
Combating Sand and Dust Storms (Pitkanen-Brunnsberg, 2019). Aligned with the UN Coalition's objectives, the World Meteorological Organization (WMO) Sand and Dust Storm-Warning Advisory and Assessment System (SDS-WAS; WMO, 2015) searches to enhance the ability of countries to deliver timely and good quality sand and dust storm forecasts, observations, information and knowledge to users through an international hub of research and operational communities (Terradellas et al., 2015, Basart et al., 2019). Monitoring and forecasting are pillars for the deployment of Early Warning
Systems (EWS) that are the most effective tool to mitigate the impacts of natural hazards such as SDS.

Winters from 2020 to 2022 were characterized by an unusually high frequency of intense and long-lasting dust events impacting the western Mediterranean and Euro-Atlantic region, which merited preliminary and widely disseminated analysis by international institutions such as WMO and the European Organisation for the Exploitation of Meteorological Satellites (EUMETSAT). These institutions described some of these dust events (see Table S1 in Section S1 of the Supplement) as
"unusual", "historic" or "exceptional". In February 2020, two strong Saharan dust events affected the Canary Islands. The first one, on 4-5 February 2020, registered a maximum peak of 1000 µg·m$^{-3}$ in the Canary Islands, and reached Iceland and the Scandinavian countries (https://public.wmo.int/en/media/news/wmo-issues-airborne-dust-bulletin; last access on 18 September 2022). The second event, on 22-24 February 2020, has so far been the most intense dust event since records began in the Canary Islands (Cuevas et al., 2021). This dust intrusion produced extremely high PM in fractions below 10 µm (PM10)
hourly concentrations in the Canary Islands, which exceeded 3000 µg·m$^{-3}$, and significantly impacted aviation (closure of the eight regional airports with the cancellation of ~1000 flights), agriculture and solar energy (Cuevas et al., 2021). This dust event also affected the western half of the Iberian Peninsula. On 18 February 2021, a dust intrusion from the Sahara swept across much of southern and central Europe, turning the snow-covered mountains of the Pyrenees and Alps orange. The dust was blown away as far as Scandinavia at the end of the month. On 15-17 March 2022, a plume of Saharan dust travelled from
North Africa and across the Mediterranean into Western Europe (CAMS technical report, https://policy.atmosphere.copernicus.eu/reports/; last access on 27 July 2023), reaching UK. The dust intrusion caused record PM10 concentrations over the Spanish air quality network (as reported by https://www.eionet.europa.eu/; last access on 27 July 2023), red- and orange-coloured skies and dust deposition on the ground across Europe, especially visible on snow in the Pyrenees and Alps. All these events were the subject of news by WMO, Copernicus and EUMETSAT (see Table S1 in Section
S1 of the Supplement), and the media disseminated shocking images of the dramatic reduction in visibility in southern European cities and of people skiing on reddish-coloured snow in the Pyrenees and the Alps because of dust deposition. These dust events caused great social alarm, being sometimes associated with the current climate change scenario.



The most frequent time of the year for Saharan dust transport to the Mediterranean basin varies strongly from one region to another. Overall, we can identify three large regions (Moulin et al., 1998; Basart et al., 2009; Pey et al., 2013; Varga et al., 2014; Gkikas et al., 2016): Eastern, Central and Western Mediterranean. In the Eastern Mediterranean, dust intrusions typically occur in spring (Israelevich et al., 2012), while in the Western Mediterranean the maximum Saharan dust transport occurs

later, in summer (Salvador et al., 2014). The central Mediterranean shows a bimodal distribution with two secondary dust peaks in spring and summer, respectively (Barnaba and Gobbi, 2004; Marconi et al., 2014).

Indeed, dust emission in West Sahara (the origin of most of the dust events reaching the western Mediterranean, e.g. Gkikas et al., 2016) occurs far away from well-known cyclogenesis regions, suggesting that dust transport might be rather associated with deep North Atlantic cyclones (Flaounas et al., 2022) or upper-level cut-off lows in the western Mediterranean (Portmann

et al., 2021). Fiedler et al. (2014) reported that cyclones are related to up to 20 % of the total dust events over the Mediterranean, which are more frequent in spring and summer (see Moulin et al., 1998; Varga et al., 2014; kikas et al., 2016) coinciding with the maximum dust activity in the Mediterranean. However, considering only extreme dust events, the Mediterranean cyclone contribution may reach 70 %.

Overall, Mediterranean cyclones that form within the Mediterranean represent 61% – 85% of the total Mediterranean tracks,

the remaining ones corresponding to Atlantic cyclones crossing the Mediterranean (Lionello et al., 2016; Flaounas et al., 2022). Cyclones on the leeward side of the Atlas Mountains, also known as "Sharav cyclones" (Winstanley, 1972), and more recently as North African cyclones or Saharan cyclones are also associated with dust storms in winter (Flaounas et al., 2022; and references herein). These cyclones are usually smaller and move faster than Mediterranean-origin winter cyclones (Alpert and Ziv, 1989), do not live more than three days, and show the maximum deepening in the tracks over the western Mediterranean

and northern Algeria (Ammar et al., 2014). Their dynamics seem to be driven by the subtropical jet (Prezerakos et al., 2006).

Both southward tracks of North Atlantic cyclones and western Mediterranean cut-off lows have been linked to atmospheric blocking (e.g. Trigo et al., 2004; Nieto et al., 2007). Blocking is a large-scale quasi-stationary equivalent-barotropic high-pressure system that can persist for several days or even weeks (Woollings et al. 2018; Kautz et al., 2022, and references herein). It has long been known that atmospheric blocking interrupts the westerly flow in mid-latitudes and diverts the eastward

path of synoptic cyclones (Rex, 1950). The region with the highest frequency of winter blocking is the North Atlantic around the Greenwich meridian (Barriopedro et al., 2006; Wazneh et al., 2021, and references herein). Blocking systems over the northern North Atlantic can result in an equatorward displacement of the storm track towards the Mediterranean basin and North Africa region (Pfahl, 2014; Kautz et al., 2022).

Winter dust intrusions over the western Mediterranean have received little attention and existing studies on the associated

atmospheric circulation have mainly focused on synoptic systems during individual case studies. For example, deep North Atlantic cyclones entering into the western Mediterranean through North Morocco and Algeria were associated with an extreme dust intrusion of February 2017 over western Europe (Fernández et al., 2019; Oduber et al., 2019), whereas Bou



Karam et al. (2009) described a strong cyclogenesis over the southern side of the Atlas Mountains associated with the African dust intrusion in February 2007 over western Mediterranean. However, a comprehensive climatological study of winter dust intrusions over the western Mediterranean, including the associated large-scale and synoptic atmospheric conditions is still lacking. This situation might be a result of 1) a relatively low frequency of occurrence of cyclone-related dust events in this region and season compared with those in spring and summer; 2) the fact that in winter and spring, clouds associated with cyclones may hamper the assessment of the dust emission and transport (Schepanski et al., 2009), 3) greater soil moisture during the winter season in North Africa results in less dust mobilization (Gherboudj et al., 2015), and 4) the wind speed thresholds required to generate dust increase in winter, therefore requiring a more extreme meteorological event, such as deep upper-level troughs/cut-offs (Cowie et al., 2014). All this leads to poor knowledge of the drivers of dust transport to Europe in wintertime.

The main goal of the present study is two-fold. Firstly, to describe the winter dust events that affected the western Mediterranean over the 2020-2022 period, and to assess to what extent they were exceptional compared to those recorded in the reference (2003-2019) period. Secondly, to characterise the atmospheric circulation systems associated with these winter dust events (considering an extended region that includes the western Mediterranean and the Euro-Atlantic regions), emphasising the role of the large-scale weather patterns and blocking, which have been largely unexplored so far.

The article is structured as follows: Section 2 describes the datasets and methods used to generate the results. Section 3 analyses the results of the dust-related datasets focusing on the identification of winter dust events over western Mediterranean for the 2003-2022 period, the associated meteorological systems (i.e., weather regimes and blocking) considering an extended region (including the western Mediterranean and the Euro-Atlantic regions) and the comparison of the recent anomalous period of 2020-2022 with 2003-2019. Finally, a brief discussion and the conclusions are presented in Section 4.

## 2. Materials and methods

### 2.1 Dust events catalogue

Meridional dust transport is estimated from the Modern-Era Retrospective analysis for Research and Applications (MERRA-2; Gelaro et al., 2017). MERRA-2 uses the GEOS-5 Earth system model (Molod et al., 2015) and the three-dimensional variational data assimilation (3DVar) Gridpoint Statistical Interpolation analysis system (GSI) (Kleist et al. 2009). MERRA-2 assimilates aerosol and meteorological observations jointly within GEOS-5 (Buchard et al., 2017). The MERRA-2 dataset contains many meteorological and atmospheric composition parameters (e.g., aerosol optical depth, AOD, by aerosol components, including dust). For the assessment of the meridional transport of dust, monthly mean values of the dust column V-wind mass flow (labelled as DUPLUXV in the MERRA-2 repository) variable have been computed for each month of the extended winters (from December to March) of 2003-2022 period. The spatial domain [27-60ºN, 30°W-36°E] encompasses North Africa, the Mediterranean basin, Europe and the eastern North Atlantic.



The dust catalogue searches to identify all the winter dust events in the western Mediterranean region for 2003-2022. The identification is based on the intensity of the event and consider three categories: Moderate, Strong and Extreme (Table 1). For the identification of dust events by each of these three categories, we use the daily AOD (at 550nm) averaged over the western Mediterranean ($AOD_{avg}$) and the AOD thresholds proposed by Gkikas et al. (2016), which must persist for at least three consecutive days (Table 1). For example, a moderate dust event occurs when the daily $AOD_{avg}$ is higher than the climatological mean ($AOD_{Clim}$) plus one standard deviation (SD), the two latter being defined from all daily $AOD_{avg}$ values of the winter months (January, February and March) of the 2003-2022 period. This yields a threshold of $AOD_{Clim} + SD = 0.18$, which must be exceeded for at least three consecutive days. Dubovic et al. (2002) and Basart et al. (2009) also considered that AOD > 0.15 values are associated with dust conditions. The identification of dust events in the western Mediterranean region (i.e., the box including [35-50°N, 20°W-5°E]) is carried out by using daily AOD (at 550 nm) from the MODIS (Kaufman et al., 1997), specifically, the Combined Dark Target and Deep-Blue at 550 nm for Land and Ocean product (Sayer et al., 2013). AOD data was obtained from the MODIS Collection 6 (Levy et al., 2013) Level 3 daily composite aerosol product, available between 2003 and 2022 with a 1º x 1º spatial horizontal resolution.

Table 1: Aerosol optical depth (AOD) thresholds used to identify and classify the intensity of the dust events.

| Type of dust event | Criteria (based on Gkikas et al., 2016) | AOD thresholds |
|---|---|---|
| **Moderate** | $AOD_{Clim} + SD < AOD_{avg} < AOD_{Clim} + 2 \times SD$ during, at least, three consecutive days | $0.18 < AOD_{avg} \leq 0.23$ |
| **Strong** | $AOD_{Clim} + 2 \times SD < AOD_{avg} < AOD_{Clim} + 4 \times SD$ during, at least, three consecutive days | $0.23 < AOD_{avg} \leq 0.33$ |
| **Extreme** | $AOD_{avg} > AOD_{Clim} + 4 \times SD$ during, at least, three consecutive days | $AOD_{avg} > 0.33$ |

The dust profile assessment is based on the Vertical Feature Mask (VFM) aerosol product obtained from the Cloud-Aerosol Lidar with Orthogonal Polarization (CALIOP; Stephens et al., 2002; Winker et al., 2009). CALIOP is an active sensor measuring the backscatter signal at 532 and 1064 nm and the polarization at 532 nm (Winker et al., 2009). The identification of cloud and aerosol layers within the atmosphere (Vaughan et al., 2009) is made through the cloud aerosol discrimination (CAD) algorithm (Liu et al., 2009; Kim et al., 2018). The VFM product aerosol subtyping algorithm distinguishes between tropospheric and stratospheric aerosols. It considers seven primary aerosol types: clean marine, dust, polluted continental, clean continental, polluted dust, smoke, and dusty marine (Liu et al., 2019). Here, we use the available Lidar Level 2.5km VFM product from NASA's Atmospheric Science Data Center (ASDC), which includes Version 4.2 (2007-2021) and Version



3.1 (for year 2022). The aerosol profile products are generated at a horizontal resolution of 5 km (https://www-calipso.larc.nasa.gov/resources/calipso_users_guide/qs/cal_lid_l2_all_v4-10.php; last access: 12 June 2023), while the vertical resolution varies from 60 to 180 m depending on the altitude range and the parameter. For the present analysis, the CALIOP VFM aerosol profile from 2007 to 2022 has been reduced to 7 vertical ranges, namely, [0-1km), [1-2km), [2-4km), [4-6km), [6-8km), [8-10km) and [>10km).

## 2.2 Atmospheric drivers

The analysis of the atmospheric drivers associated with winter dust events over our study region is based on the National Centre for Environmental Prediction / National Centre for Atmospheric Research (NCEP/NCAR) Reanalysis (Kalnay et al., 1996). The NCEP/NCAR Reanalysis is one of the state-of-the-art atmospheric reanalysis systems providing consistent information from 1948 to the present. For the present study, six-hourly and daily mean fields at different pressure levels and 2.5° x 2.5° spatial resolution have been used to characterise the large-scale atmospheric conditions associated with the dust events identified in Section 2.1.

The analysis is done in two parts. First, an identification of recurrent weather patterns associated with dust events, and secondly, the analysis of specific large-scale phenomena, including the North Atlantic jet stream and blocking events. To categorise the large-scale atmospheric circulation in a limited number of recurrent weather regimes (WRs) we have followed the extended year-round classification of Grams et al. (2017), which uses seven different WRs, reflecting the preferred flow patterns within the whole spectrum of states of the atmospheric circulation in the Euro-Atlantic region. A WR index is calculated every day and for each WR as the anomaly of the projection normalized by the standard deviation (Michel and Rivière 2011). To be attributed to a specific WR, the respective index must be greater than 1 and all other WR indices of that day. The seven WRs are defined as follows: the Zonal regime (ZO) and its variant Atlantic Trough (AT); the related Scandinavian Trough (ScTr) and Atlantic Ridge (AR) regimes; European Blocking (EuBL) and Scandinavian Blocking (ScBL); and Greenland Blocking (GL). For further information on WR characteristics see Beerli and Grams (2019).

A similar classification approach has been applied to the subset of dust days in order explore the diversity of large-scale patterns associated with dust intrusions in the western Mediterranean. The analysis of favourable patterns for dust intrusions is based on the K-means clustering algorithm (e.g., Wilks, 2011), which classifies data into a predefined number of groups, which maximise (minimise) the inter-cluster (intra-cluster) Euclidean distance. We have followed the same methodology used by Alonso-Pérez et al. (2011) (see details of the methodology therein) to obtain the four main daily patterns of geopotential height (Z) anomalies associated with dust events. Daily anomalies are defined as the difference between the daily value and the mean of the corresponding calendar month for the 1991–2020 standard reference period. The method classifies all days with a dust event in one of the four weather patterns. The clustering has been applied to daily Z500 anomalies for the entire winter season (January-to-March) over the [20-70° N, 65° W-25° E] domain and the periods 2003-2019 and 2020-2022,



separately. The selected number of clusters was determined as a compromise, considering the relatively low number of winter dust days, particularly for the short recent period of 2020-2022.

On the other hand, the localisation of the North Atlantic eddy-driven jet stream and blocking patterns is performed on a daily basis, following the detection algorithms of Barriopedro et al. (2022) and Woollings et al. (2018), respectively. The retrieved fields (daily maps of binary fields) allow us to quantify the local frequency of jet and blocking occurrences during a given time interval or subsample of days. They are herein used to identify the favourable conditions (jet configurations and blocking activity) for the occurrence of dust events during the "normal" period of 2003-2019 and the recent anomalous period of 2020-2022. We note that the jet stream is detected every day of the analysed period, whereas blocking may or not be present on a given day. Therefore, the blocking catalogue is also employed to distinguish dust days with and without blocking, which allows us to assess the relative importance of blocking and other large-scale atmospheric conditions associated with dust.

The jet stream is diagnosed from 10-day low-pass filtered fields of the 925-700 hPa averaged zonal wind (see more details in Barriopedro et al., 2022). The latitudinal position of the jet is inferred from the daily meridional profile of the zonal wind at each longitude, which has been previously smoothed by averaging the zonal wind over a longitudinal sector of 60º width centred at that longitude. The latitudinal position of the jet is the latitude of the zonal wind maximum in the meridional profile. The poleward and equatorward boundaries of the jet correspond to the latitudes at each side of the jet latitude where the zonal wind maximum decreases by a given amount (defined as half the difference between the zonal wind peak and the meridional mean of the zonal wind over 15-75º N). All latitudes between the poleward and equatorward flanks of the jet identify the jet structure (location and width of the zonal wind maxima) at that longitude. When applied to all longitudes of the Euro-Atlantic sector, the method provides a 2-D daily identification of the North Atlantic jet.

Blocking patterns are identified from Z daily fields at 500 hPa (Z500), following the meridional gradient reversal method described in Woollings et al (2018). For each day d and longitude $\lambda$, meridional Z500 gradients are computed to the north and south of a given latitude $\varphi$ as follows:

$$GHGN(\lambda, \phi, d) = \frac{Z500(\lambda, \phi+\Delta, d) - Z500(\lambda, \phi, d)}{\Delta} \quad (1a)$$

$$GHGS(\lambda, \phi, d) = \frac{Z500(\lambda, \phi, d) - Z500(\lambda, \phi-\Delta, d)}{\Delta} \quad (1b)$$

$$GHGS2(\lambda, \phi, d) = \frac{Z500(\lambda, \phi-\Delta, d) - Z500(\lambda, \phi-2\Delta, d)}{\Delta} \quad (1c)$$

where $45° < \varphi < 70°$ N and $\Delta = 15°$ latitude. GHGS is employed to detect high-low blocking patterns, whereas GHGN and GHGS2 aim to filter out other flow reversals unrelated to block. The meridional gradients in Eq. (1) are averaged over $\Delta/2$ in longitude, denoted by the overbars in Eq. (2):

$$\overline{GHGN}(\lambda, \phi, d) < -10 \, m/º \quad (2a)$$

$$\overline{GHGS}(\lambda, \phi, d) > 0 \, m/º \quad (2b)$$



$$\overline{GHGS2}(\lambda, \phi, d) < -5\,m/º \quad (2c)$$

Daily blocking-like structures are identified as spatial clusters of contiguous grid points satisfying the Eq. (2) conditions and covering a minimum areal extension of 500,000 km$^2$. Finally, the blocked areas detected in consecutive days are required to overlap (non-zero overlapping) and persist for at least five days to be considered a blocking event.

## 3. Results

The results are structured in two sections, which aim to describe the climatological characteristics of wintertime dust events in the western Mediterranean, paying special attention to the anomalous 2020-2022 period (Section 3.1) and to characterise the atmospheric circulation associated with these events (Section 3.2).

10 **3.1 Assessment of African dust transport into the western Mediterranean basin in wintertime**

We start analysing the time series of meridional dust transport over the western Mediterranean (i.e. [35-50ºN, 20ºW-5ºE]) as inferred from the spatially averaged monthly mean dust column V-wind mass flux of the MERRA-2 reanalysis (Figure 1 (a)). The western Mediterranean shows significant intraseasonal and interannual variability of meridional dust transport. The highest (lowest) values tend to occur in late winter (January) (Figure 1 (a)). The meridional transport of mineral dust towards our geographic domain displayed a pronounced increase in the 2020-2022 period, registering the highest values of the series, which correspond to the months of February and March 2021, and March 2022. These results confirm the increase of winter dust transport towards the western Mediterranean in the 2020-2022 period compared to those recorded in the previous observational (2003-2019) period.





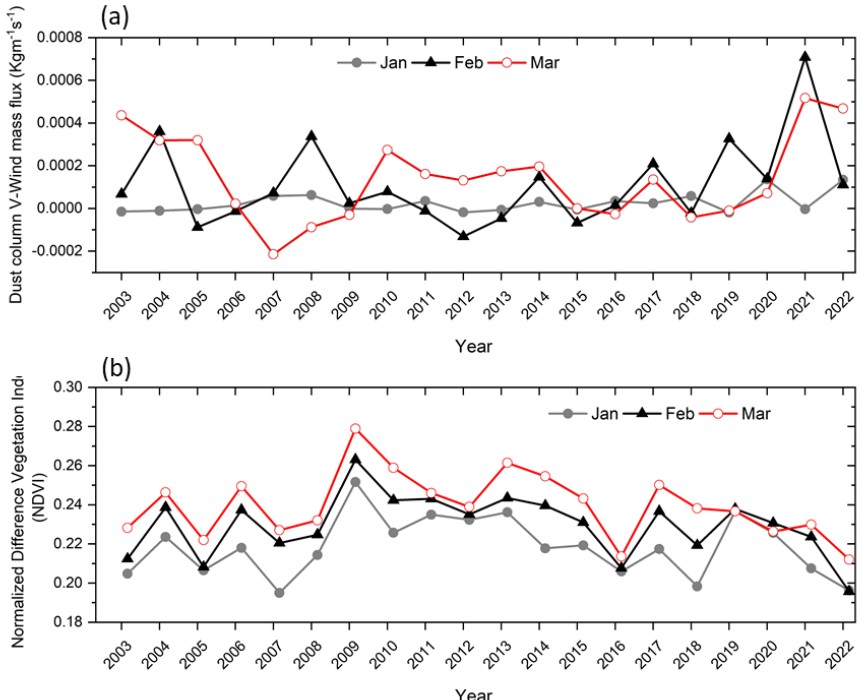

**Figure 1: Time series of averaged region-monthly mean for each winter month (i.e., January, February and March) and the 2003-2022 period. a) Dust column V-wind mass flux (kg m$^{-1}$ s$^{-1}$) based on MERRA-2 in the geographic domain [35-50ºN, 20ºW-5ºE]. b) Normalized Difference Vegetation Index (NDVI) based on MODIS over the northernmost strip of the Maghreb [32-36.5°N, 7°W-10°E].**

During the climatological 2003-2019 period, the spatial patterns of monthly mean dust column V-wind mass flux show that wintertime Saharan dust intrusions towards the Mediterranean basin take place preferentially over the eastern Mediterranean through the coasts of Libya and western Egypt, affecting mainly Greece and southern Italy (Figures 2 (a-d)). Moreover, dust intrusions increase in intensity and areal extent as winter progresses, reaching a maximum in March. Overall, the impact on the western Mediterranean is very small, which agrees with the results reported in previous studies (e.g., Vargas et al., 2014; Gkikas et al., 2015; Gavrouzou et al. 2021). During 2020-2022, the spatial patterns changed significantly since dust intrusions occurred further west (through Morocco and Algeria), fully impacting the western Mediterranean region due to dust transport from the western Sahara (Figures 2 (e-h)). Dust transport through the usual corridors of the eastern Mediterranean almost disappeared during this anomalous period, which represents an extraordinary situation. Based on these results we have selected an appropriate study area to detect dust intrusions in the western Mediterranean. This study area is indicated in Figure 2 by a rectangle with a white perimeter. The southern parallel has been chosen far enough from North Africa to avoid the detection of short-range, low-intensity dust intrusions associated with mesoscale processes. The northernmost boundary has not been





chosen too far north so that the averaged dust content in the geographical domain is not very diluted. Furthermore, given that dust intrusions in the western Mediterranean are negligible in December (Figures 2 (a), (e)), the analysis of the results are discussed considering the winter months, separately.

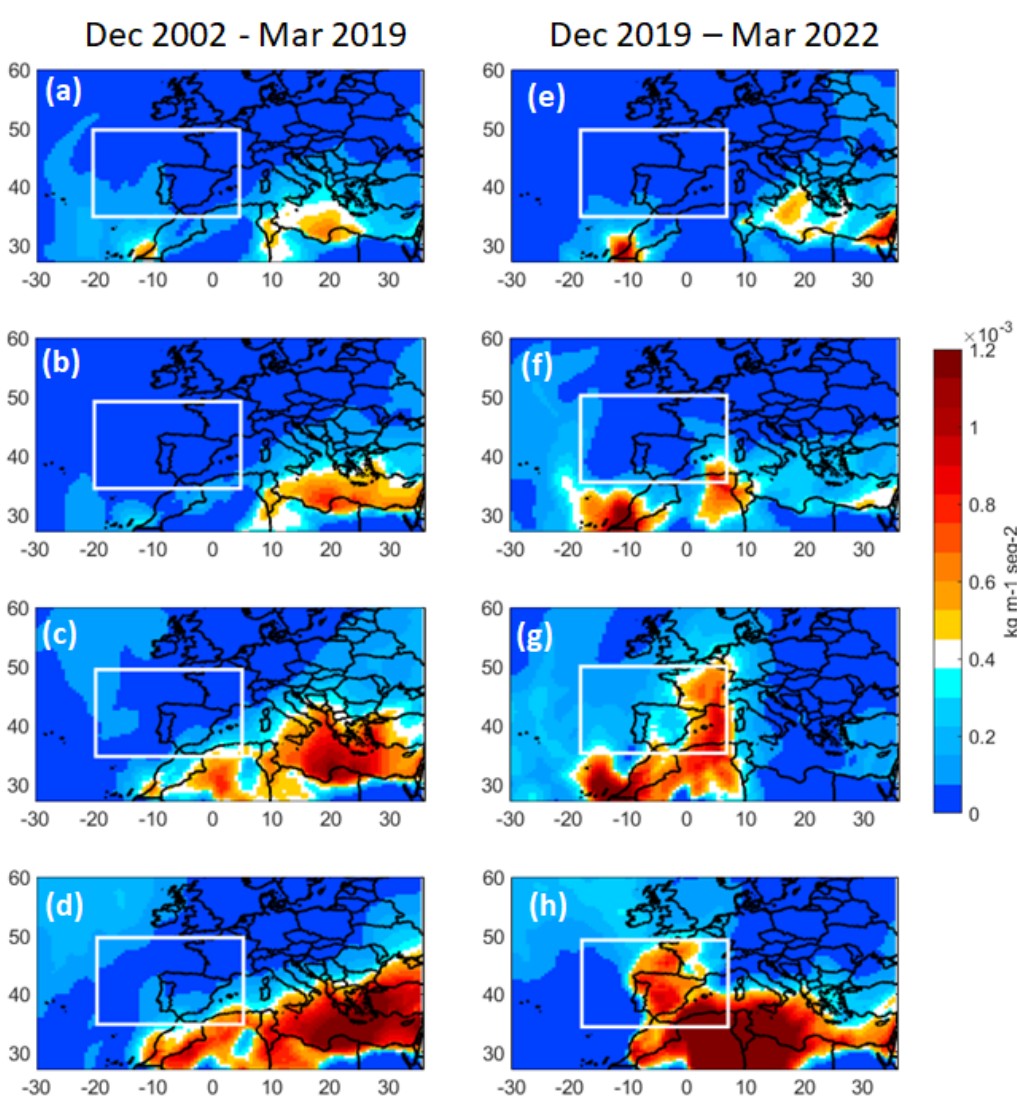

**Figure 2:  Mean dust column V-wind mass flux (DUPLUXV; kg m$^{-1}$ s$^{-1}$) for: (a, e) December, (b, f) January, (c, g) February, and (d, h) March of the 2002-2019 period (a-d) and 2020-2022 period (e-h). The white box delimits the spatial domain [35-50ºN, 20ºW-5ºE] considered for the study of dust intrusions in western Mediterranean. Results are based on MERRA-2 reanalysis.**



Because of the limitations of global and regional aerosol reanalyses to capture the intensity of extreme events (e.g., Buchard et al., 2017; Mytilinaios et al., 2023), the identification of dust events in the study region for 2003-2022 follows the methodology described in Section 2.1, which relies on satellite MODIS aerosol products. To proof the consistency of the

results between MERRA-2 and MODIS, a comparison of these datasets is included in Section S2 of the Supplement. The mean absolute difference between the daily AOD time series of MODIS and MERRA-2 reanalysis in our study region (which comprise a total of 1796 days, see Figure S10 in Section S3 of the Supplement) is very small (<< 0.01) and the Pearson correlation coefficient is 0.87 (or 0.77 when using only dust days). A drawback of the MODIS dataset might be the presence of clouds within some atmospheric systems, which prevent satellite dust-AOD measurements. This may seem especially

inconvenient in winter when the strongest cyclones are expected to develop over the Mediterranean Sea. However, we have verified that this approach, already used in previous studies (e.g., Moulin et al. 1998; Barnaba and Gobi, 2004; Gkikas et al, 2016) is robust and reliable. Following Gkikas et al. (2016) we have also compared the MODIS $AOD_{avg}$ with five AERONET European stations (Giles et al., 2019; see details in Section S2 of the Supplement). The correlation for January to March period and from 2003 to 2022, using data only when there is a dust event, is in the 0.41-0.67 range (see Table S2 in Section S2 of the

Supplement), which is in general consistent with similar comparisons performed by Gkikas et al. (2016) and Gavrouzou et al. (2021).

In the period 2003-2022, and according to the AOD thresholds specified in Table 1, we have identified a total of 32 dust events (1.7 dust-events/year) in our study region. These events are required to have a daily mean AOD over the geographic domain of at least 0.18, and a duration of three or more consecutive days. Depending on their mean intensity (i.e., the daily mean AOD

averaged over all days of the event) dust events have been classified as moderate, strong, or extreme. Note that dust events lasting less than 3 days, or with daily mean AOD lower than 0.18 are not considered in this study.

The main characteristics of the dust events are summarised in Figure 3, including the number of dust days and its average AOD for each winter month and year of the 2003-2022 period. This information is completed with Table 2, which shows the number of dust days and events for each calendar month and analysed period (2003-2019 and 2020-2022). The occurrence of

dust events in the month of January is very sporadic as only two events have been recorded (one of them, labelled as strong, in the anomalous 2020-2022 period). In February the occurrence of dust events increases considerably as compared to January, but it remains very irregular over the analysed period, including a period of six consecutive years (2010-2015) without any event. Like in January, the intensity of February events tends to be moderate (the only one labelled as extreme occurred in 2017). The anomalous 2020-2022 period displayed an outstanding increase in the frequency of February dust intrusions, both

in terms of events and days, which were approximately five and three times higher than those registered in the reference 2003-2019 period (Table 2). Indeed, five out of the 11 dust events recorded in February since 2003 have occurred in the 2020-2022 period, including one strong event (Figure 3). As expected, March, halfway between late winter and early spring, is the winter month with the highest frequency of dust events in western Mediterranean. After a quiet period of four years (2016-2019), the



anomalous 2020-2022 period registered five dust events, including two extreme and two strong episodes. When the whole winter is considered, the recent 2020-2022 period experienced an outstanding (eight-fold increase) in both the mean frequency of winter dust events and days as compared to 2003-2019 (Table 2).

In summary, every winter of the 2020-2022 period reported dust intrusions in at least two of the three winter months, more than 15 dust days per winter and at least one strong/extreme event. Overall, the analyses (Figure 3 and Table 2) confirm that the 2020-2022 period was the most active one in terms of frequency, duration and intensity of dust events in the western Mediterranean since records began. Compared to previous years, many of those dust events were long-lasting (>8 days) and intense (AOD> 0.25) episodes. This increase of the occurrence of intense dust events in the western Mediterranean during 202-2022 is statistically significantly different from the reference 2003-2019 period (p-value of ~$10^{-6}$) as derived from the Wilcoxon rank sum test. In particular, the 14-day dust event of March 2022, with an AOD mean of 0.4 constitutes a milestone in dust transport from North Africa to the western Mediterranean region. This event broke all records in our study region in terms of duration, intensity, and affected area, and affected air quality in western and central Europe with high PM10 values (50-100 µg·m$^{-3}$) even on high mountains such as the Pyrenees and the Alps. The results agree with those inferred from MERRA-2 and support the selected separation of periods. For the dust events of the 2003-2022 period, the AOD fields from MERRA-2 are very similar to those of MODIS (Figure S9 in Section S2 of Supplement). Overall, AOD values are slightly lower in MERRA-2 than in MODIS (by ~0.02; Figure S10 in Section S2 of Supplement), and hence MERRA-2 tends to underestimate the intensity of dust intrusions in our study region.





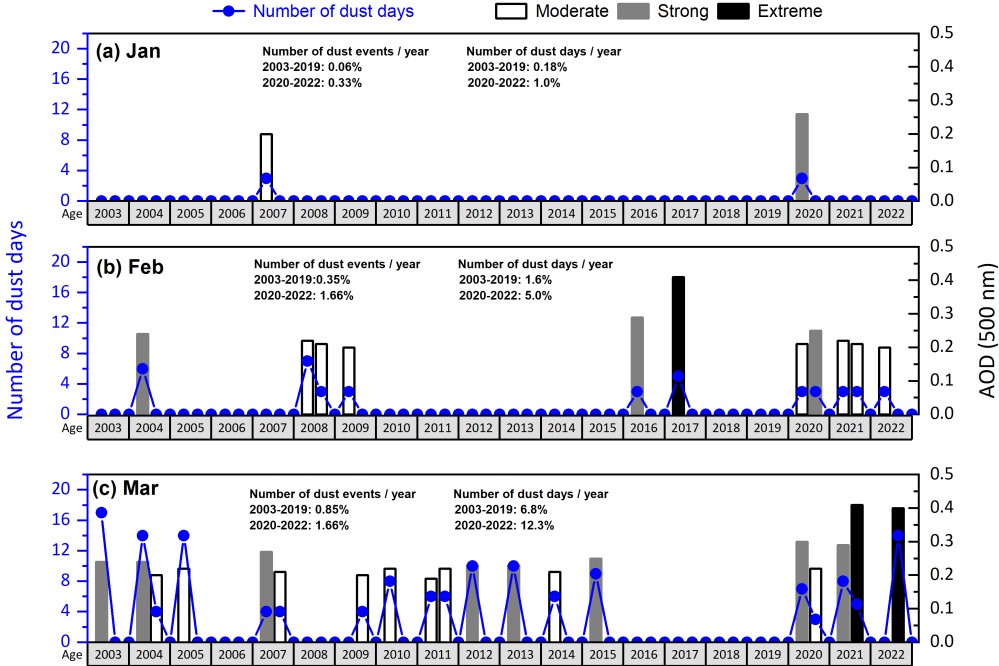

**Figure 3: Monthly time series (2003-2022) with the number of dust days (blue dots and lines; left y-axis) and the AOD at 550 nm averaged for all dust days in the month (bars; right y-axis) for: (a) January, (b) February, and (c) March. The colour of the bars indicates the type of the dust event (white: moderate (0.18 < $AOD_{avg}$ ≤ 0.23), grey: strong (0.23 < $AOD_{avg}$ ≤ 0.33) and black: extreme ($AOD_{avg}$ > 0.33)). Results are based on MODIS aerosol retrievals.**

**Table 2: Mean number of dust events per year, and mean number of dust days per year for each winter month of the 2003-2019 and 2020-2022 periods. Results are based on MODIS aerosol retrievals.**



|  | 2003-2019 | | | 2020-2022 | | |
|---|---|---|---|---|---|---|
|  | Jan | Feb | Mar | Jan | Feb | Mar |
| Dust events (# yr$^{-1}$) | 0.06 | 0.35 | 0.82 | 0.33 | 1.66 | 1.66 |
| Dust days (# yr$^{-1}$) | 0.18 | 1.6 | 6.8 | 1.0 | 5.0 | 12.3 |

According to the dust climatology of Gkikas et al. (2016), dust layers in the Mediterranean are normally observed between 2 and 6 km height during summer, while in winter they are mainly detected below 3 km and over the central and eastern Mediterranean Sea. Here, we have calculated the mean percentage of pure dust in seven tropospheric layers (between 0 and 10 km) of our geographical region based on the aerosol vertical profiles from CALIOP (see Section 2.1), which are available for the anomalous (2020-2022) and a more recent reference period (2007-2019). For the period 2007-2019, the results show slightly higher percentages of pure dust at higher altitudes (2-4 km) than those reported by Gkikas et al. (2016) (< 2km) (Table 3). The percentages of pure dust increase significantly in the three months of the recent 2020-2022 period, but especially in February and March. In this last month, there is a non-negligible percentage of pure mineral dust in an upper layer, between 4 and 6 km of altitude. These results suggest that the enhanced intensity of winter dust intrusions during 2020-2022 could have been mediated by convective processes lifting mineral dust from the desert to heights higher than usual. The role of synoptic to large-scale systems is addressed in the following section. Finally, other factors could have contributed to exacerbate the frequency, duration and/or intensity of recent dust events. In the winter of 2021-2022, an intense drought was recorded (JRC, 2022), which affected the arable areas of northern Morocco and Algeria. Indeed, during the winter months of 2022, the monthly mean series of the Normalized Difference Vegetation Index (NDVI), which accounts for vegetation cover and activity, recorded some of the lowest values of 2003-2022 over the northernmost strip of the Maghreb (see Figure 1 (b)).



**Table 3: Percentage of pure dust with respect to the total available aerosol observations per vertical layer for the dust events of 2007-2019 and 2020-2022, as inferred from the CALIOP Vertical Feature Mask (VFM) product.**

|  | 2007-2019 | | | 2020-2022 | | |
| --- | --- | --- | --- | --- | --- | --- |
|  | **Jan** | **Feb** | **Mar** | **Jan** | **Feb** | **Mar** |
| **0-1 km** | 0.02 | 0.03 | 0.03 | 0.02 | 0.05 | 0.04 |
| **1-2 km** | 0.02 | 0.04 | 0.04 | 0.03 | 0.08 | 0.06 |
| **2-4 km** | 0.02 | 0.05 | 0.05 | 0.04 | 0.12 | 0.14 |
| **4-6 km** | 0.02 | 0.03 | 0.03 | 0.02 | 0.03 | 0.09 |
| **6-8 km** | 0.02 | 0.02 | 0.03 | 0.02 | 0.02 | 0.04 |
| **8-10 km** | 0.02 | 0.01 | 0.02 | 0.02 | 0.02 | 0.02 |
| **> 10 km** | 0.02 | 0.01 | 0.02 | 0.02 | 0.02 | 0.01 |

## 3.2 Atmospheric drivers of dust events over the Western Mediterranean and Euro-Atlantic in wintertime

This section describes the synoptic to large-scale atmospheric patterns associated with dust days in the western Mediterranean. The diversity of atmospheric configurations will be assessed through statistical tools (weather types) and automatic algorithms tracking specific phenomena (blocking and jet stream) over an extended region that includes the western Mediterranean and the Euro-Atlantic, as described in Section 2.2. The meteorological systems at synoptic scales (cyclones, cut-off lows, etc.) have been identified subjectively by visual inspection of the dust events, which has also allowed us to confirm the results of the objective analyses. Section S1 of the Supplement provides a more detailed analysis of the three dust episodes labelled as "extreme" in the 2003-2022 period (see Figure 3), which occurred on 20-24 February 2017, 27-31 March 2021, and 15-31 March 2022. A common feature of the three episodes is the presence of at least one very deep cut-off low (visible at 100 hPa) between the Canary Islands and the Iberian Peninsula. These cut-off lows lift dust from sources located in Morocco and Algeria and transport it to western Europe. In the first two cases (20-24 February 2017, and 27-31 March 2021) we identified a single cut-off low to the west of the Atlas Mountains, whereas in the third case (15-31 March 2022) four concatenated cut-off lows moved eastwards from the Atlantic to the Mediterranean coasts of Morocco and Algeria. The three cases showed large-scale





### 3.2.1. Weather regimes

Figure 4 shows the four Z500 anomaly patterns obtained from the K-means clustering of the winter dust days and the 2003

2019 2020-2022 periods. The most frequent patterns in both periods (Cluster C#1; Figures 4 (a), (e)) are characterised by a meridional pressure dipole with positive Z500 anomalies between the British Isles and Scandinavia and negative Z500 anomalies to the south, which corresponds to a European Rex-like block (Rex et al., 1950; Sousa et al., 2021). This favourable configuration was slightly more frequent in the reference 2003-2019 period (47.6 %) than in the 2020-2022 period (40.0 %), but it was more intense in the latter. The second most frequent pattern (Cluster C#2; Figures 2 (b), (f)) shows a zonal dipole,

with a widespread negative Z500 anomaly from southern Greenland to western Europe and a positive Z500 anomaly over eastern Europe, which resembles the spatial pattern of Ural blocks (e.g., Barriopedro et al., 2006). This pattern is also broadly identified in the two subperiods, with similar frequencies of occurrence (less than 30%). The third most recurrent cluster (C#3, ~20%; Figures 4 (c), (g)) shows a north-south oriented anomalous ridge over the North Atlantic flanked by centres of negative Z500 anomalies, like omega-shaped North Atlantic blocks (e.g., Sousa et al., 2021). Cluster C#4, which resembles a Rex-like

block over the North Atlantic (Figures 4 (d), (h)), occurs with very low frequency in the reference 2003-2019 period (<5%), but it was quite common in the recent 2020-2022 period (20%). Overall, C#3 and C#4 patterns showed significant subtropical ramifications in the 2020-2022 period, as compared to the 2003-2019 period, and more marked signatures of subtropical cut-off lows.

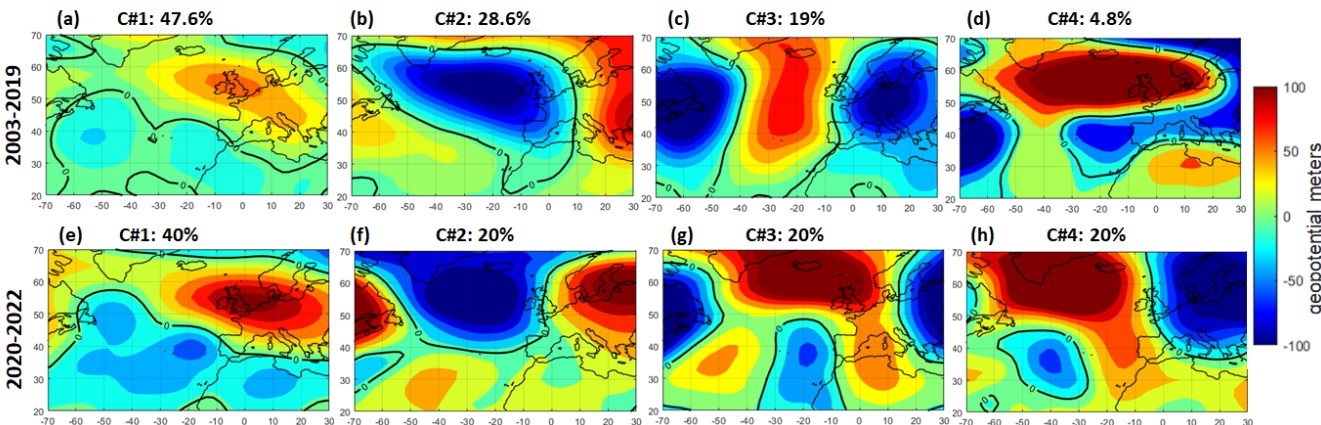

**Figure 4: Four clusters (C#1 to C#4) of Z500 anomalies (in m) for the winter dust days of the reference 2003-2019 (a-d) and 2020-2022 (e-h) period. The relative frequency of each cluster (in % with respect to the total number of dust days of each period) is indicated on the top. Geopotential height fields have been obtained from NCEP/NCAR reanalysis.**





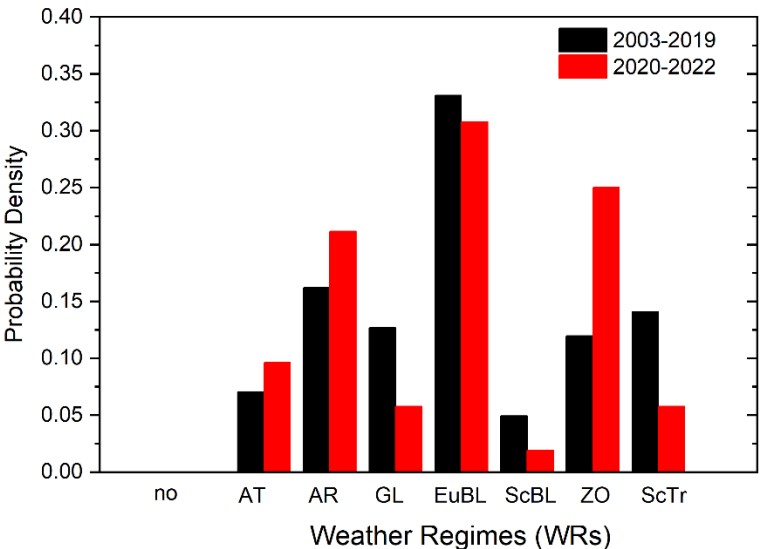

**Figure 5. Probability of occurrence of weather regimes (WRs) during dust days of 2003-2019 (black bars) and 2020-2022 (red bars). The following acronyms are used. no: no regime; AT: Atlantic trough; AR: Atlantic ridge; GL: Greenland blocking; EuBL: European blocking; ScBL: Scandinavian blocking; ZO: zonal regime and ScTr: Scandinavian trough.**

To investigate the evolution of the large-scale circulation over the Euro-Atlantic region and its relationship with the above-shown clustering patterns, we consider the seven WRs of Grams et al. (2017). Figure 5 shows the frequency density distribution of WRs for the dust days of each period. We note that none of the WRs accounts specifically for the transport of air masses from North Africa to Europe and that some WRs do not have a direct apparent correspondence with the clusters of Figure 4 (e.g., GL, ZO). Despite this, the analysis of WRs and their correspondence with the clusters of Figure 4 reveal the following results:

- The WR with a higher probability of occurrence during dust days is the EuBL WR (Figure 5), which is consistent with the high frequency of occurrence of the C#1 clusters in the two periods (Figures 4a, e).

- The AR is the second most likely WR during dust days (Figure 5), showing a small increase in probability of occurrence during the 2020-2022 period. The increase in the probability of AT during the dust days of the 2020-2022 period could also be explained by clusters C#3 and C#4 (Figures 4g, h)., which display negative Z500 anomalies over the North Atlantic, and account for a large fraction (40%) of dust days of that period.



- The frequency of the ZO WR increased (by a factor of two) during dust days of the 2020-2022 period with respect to 2003-2019. This WR does not have a direct correspondence with any of the clusters of Figure 4 but is consistent with the increase in subtropical ridges reported in Section 3.2.2, which is partially accounted for by the clusters C#3 and C#4 of the 2020-2022 period.

- A decrease in the probability of ScTr WR is observed in the second period, consistent with the lower frequency shown by cluster C#2 in the 2020-2022 period compared to that of the same cluster in the normal period 2003 -2019.

## 3.2.2. Meteorological drivers

In this section, we analyse the weather systems associated with dust intrusions in the western Mediterranean following the methods described in Section 2.2. First, we carried out a subjective analysis of the Z fields at different pressure levels (from 925 to 100 hPa) for each day of the 32 winter dust events of 2003-2022. The results revealed the following common signatures: 1) synoptic systems: negative Z anomalies (typically < -80 m) at all levels from 925 to 200 hPa located in the subtropical region of the eastern North Atlantic or in North-western Africa, which are identified as cut-off lows; and 2) large-scale systems: relatively extensive regions of positive Z anomalies (typically > 100 m at 500 hPa) in mid or high latitudes. The latter can occur under a variety of flow configurations, as confirmed by Figures 4 and 5. A substantial number of cases resembles the cluster C#1 patterns (Figures 4a, e) or EuBL WR, with positive Z anomalies located to the north of the cut-off low, therefore forming a dipole-like block (Rex, 1950; Barriopedro et al. 2010). A second group, like AR WR, reveals positive Z anomalies at comparatively lower latitudes, which are more characteristic of subtropical ridges (Sousa et al., 2021) or low-latitude blocks (Davini et al., 2012) over the eastern North Atlantic or western Mediterranean. The next sections characterise the synoptic (cut-off lows) and large-scale (high-pressure) systems associated with dust days. We also describe the jet stream, since these weather systems are often associated with substantial departures of the westerlies and the storm track activity (Trigo et al., 2004). The analysis relies on composites for the 2003-2019 and 2020-2022 period, separately, to emphasise distinctive features of the recent anomalous periods

### 3.2.2.1. Main synoptic features

For all dust events, the cross-section of Z anomaly shows a robust structure with negative Z anomalies throughout the troposphere between 925 and 200 hPa (not shown). The Z anomalies (in absolute value) at 200 hPa are in all cases larger than those found at 500 hPa, consistent with the typical signatures of upper-level cut-off lows (Nieto et al., 2005, and references herein). Indeed, more than half of the winter dust events of the 2003-2019 reference period concurred with a cut-off low. This relationship strengthened during the anomalous 2020-2022 period, particularly in March, when all dust events were accompanied by cut-off lows. During dust days, cut-off lows are normally located between the Canary Islands and the Iberian Peninsula, activating the dust sources identified over Morocco and Algeria and transporting dust northwards (see Section S1 in the Supplement). Noticeable Z anomalies were also found in the lower stratosphere (100 hPa) during at least 50 % of the 2003-2019 cases and all dust events of the anomalous 2020-2022 period. The deeper and stronger Z signatures of the 2020-





2022 events are consistent with detecting dust at high heights. The negative Z anomalies at the lowest (925 hPa) level are weaker than at higher levels (not shown) but intense enough to generate surface winds exceeding the wind speed threshold for dust mobilization in the west Sahara, which is between 5 and 12.5 m·s⁻¹ (Helgren and Prospero, 1987). In a few cases, a near-surface cyclone can be initiated by the circulation from aloft during the later phases of the cut-off low.

For some dust events the centres of these cut-off lows remained stationary over the Atlantic coast of Morocco during ~3-4 days, mobilizing dust on its eastern flank (see some examples in Figure S1 in Section S1 of the Supplement), before being absorbed by the general circulation. However, stationary cut-off lows were relatively uncommon; in most cases, cut-off lows moved eastward from the subtropical North Atlantic to the western Mediterranean. Transient cut-off lows can comprise two or more successive cut-off lows. This situation was identified in at least one third of the 2003-2019 dust events, but in all

March dust events of the 2020-2022 period. The maximum number of concatenated cut-off lows in a dust event was four, registered in the historical episode of 15-30 March 2022 (see case analysis #3 in Section S1.3 of the Supplement). Moreover, the cut-off lows detected during dust events tend to be long-lasting, in agreement with the characteristic time scale (~1 week) of the accompanying high-pressure systems (Barriopedro et al, 2006), which act to weaken the mean zonal flow and slow down the reabsorption of the cut-off lows by the westerlies.

Apart from this, we do not identify distinctive features of the cut-off lows associated with dust events. Indeed, their characteristics are consistent with those reported in previous climatological studies (e.g. Nieto et al., 2005, 2007): 1) winter cut-off low occurrence is mostly related to blocking events; 2) winter cut-off lows of the European sector tend to occur in northern Africa and southern Europe; 3) cut-off lows often last 3–5 days, and very few last more than five days; and 4) more than 80% of cut-off lows are highly transient during their life cycle.

**3.2.2.2. Main large-scale features**

To describe the main large-scale features associated with dust days, we computed the local frequency of blocking and jet stream occurrence for all winter dust days and compared them with their climatological frequencies (regardless of dust occurrence). Figures 6 (a) and (c) show blocking and jet frequency exceedances above the climatology during dust days, respectively. The analysis focuses on February-March (FM) winter months of the reference (2003-2019) and anomalous (2020-

2022) period, although we obtain similar results when the entire winter season is considered. Significance is assessed with a bootstrapping of 500 members, each one derived from a random choice of winter days of the same size as the number of dust days. Dust days are associated with enhanced blocking activity over a latitudinal band centred at mid latitudes (~50-60º N) and extending from the eastern Atlantic Ocean to Scandinavia. Therein, the probability of blocking occurrence more than doubles during dust days as compared to the climatology. The North Atlantic jet is shifted poleward, with preferred locations

over high latitudes and reduced frequencies over typical jet latitudes, which is in agreement with the obstruction of the





westerlies by the blocking action and its diverting effect on the jet (Trigo et al. 2004; Barriopedro et al. 2010; Sousa et al. 2017). Therefore, European blocking with poleward shifted jets represent a favourable configuration for dust occurrence.

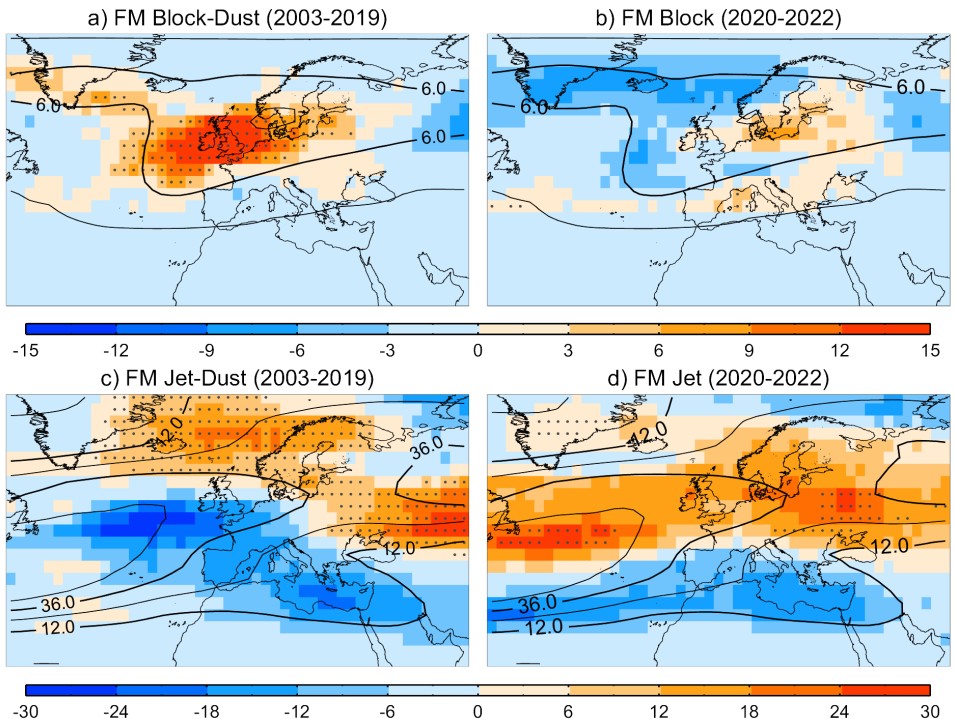

**Figure 6. Blocking (a,b) and jet stream (c,d) frequency anomaly during dust days (in our study region) in February-March (FM) of the reference 2003-2019 (a,c) and 2020-2022 (b,d) period. Frequencies are expressed in percentage of dust days of the February-March period and anomalies (colour shading) are defined with respect to the expected frequency of occurrence (contours). Dots denote significant positive anomalies at p<0.05 (i.e. grid points where local blocking and jet frequency is significantly higher than expected), as inferred from a 500-trial bootstrap. The expected frequency of occurrence is defined as the mean of the 500 random members, all with the same sample size as the number of considered days but random dates of occurrence. Geopotential height fields have been obtained from NCEP/NCAR reanalysis.**

Figures 6 (b) and (d) show the anomalies in blocking and jet frequency for FM 2020-2022. Overall, blocking activity was not significantly higher than the climatology, suggesting that the anomalous frequency of dust days in 2020-2022 cannot be fully explained by a corresponding blocking increase over the favourable region for dust intrusions. However, the spatial pattern hinders important intra-seasonal differences (not shown). In February 2020-2022, blocking activity was almost suppressed



over central Europe, but it was recurrent at lower latitudes, resulting in a significant increase over the Mediterranean (also evident on seasonal scales). A high frequency of low-latitude blocks is commonly associated with subtropical ridges, which can temporarily exhibit meridional gradient reversals (Sousa et al., 2021). These high-pressure low-latitude structures reinforce the zonal wind at their poleward flanks, favouring a mid-latitude intensification of the jet, rather than its poleward migration (Sousa et al. 2017), as in February 2020-2022 (not shown).

Differently, March 2020-2022 was characterized by poleward jets and enhanced blocking over the climatological region of occurrence, like the canonical pattern associated with dust (and the cluster C#1 of Figure 4). Despite this, the association between blocking and dust days weakened as compared to the climatology (i.e., the percentage of dust days concurring with blocks was lower in March 2020-2022 than in 2003-2019). To quantify this better, we computed the percentage of dust days when a block was detected over the (25° W-30º E) domain. During the reference period (2003-2019), more than 65% of the dust days in March coincided with block, which is significantly higher than that expected from the climatology (binomial test, p<0.01). This conditional probability decreased to ~50% during March 2020-2022 (p>0.1). A similar weakened block-dust linkage is also observed for the entire FM season of 2020-2022. Therefore, although blocking activity and jet configurations were favourable for the outstanding frequency of winter dust days in 2020-2022, they explained less exceedances than those expected from the historical record.

Several hypotheses could explain the weaker role of blocking in driving dust intrusions over the study region during the recent 2020-2022 anomalous period. After dust transport, air stagnation (i.e., weak winds and low precipitation impeding the dispersion and deposition of PM; e.g., Garrido-Pérez et al. 2018) could have favoured the maintenance of high AOD levels in the atmosphere. Although these conditions can indeed take place between the meridional flow periods associated with dust intrusion and the reestablishment of the westerlies that clean the atmosphere, we did not find an unusually high frequency of stagnant days during 2020-2022 (not shown). Therefore, we explored if dust days of the 2020-2022 period were associated with unusual blocking configurations and other (non-blocking) high-pressure systems. To do so, we computed the composites of Z500 anomalies for dust days with and without simultaneous blocking over the [25°W-30ºE] domain (Figure 7). The analysis is performed for the reference (2003-2019) and recent (2020-2022) periods separately. As expected, block-dust concurrence during the reference period shows the typical configuration of a high-low pressure dipole, with low pressures to the south of a blocking high that is fully detached from the subtropical belt (Figure 7 (a)). During the 2020-2022 period, dust-related blocks were located at comparatively eastern longitudes, and displayed higher Z500 anomalies, which extended towards the Mediterranean (Figure 7 (b)). This equatorial elongation of Z500 anomalies denotes an Omega-like blocking pattern (i.e., without complete detachment of the blocking high), for which meridional reversals are typically confined upstream and downstream of the blocking high. The distinctive structure of the 2020-2022 blocked flows becomes clear in Figure 7 (c), which shows the Z500 difference between the blocking-like patterns associated with dust days in 2020-2022 and 2003-2019.





The pattern evidences an eastern shift and equatorial elongation of 2020-2022 blocking (positive Z500 differences over continental Europe), with a deeper trough upstream of the blocking high (negative Z500 differences over the eastern Atlantic).

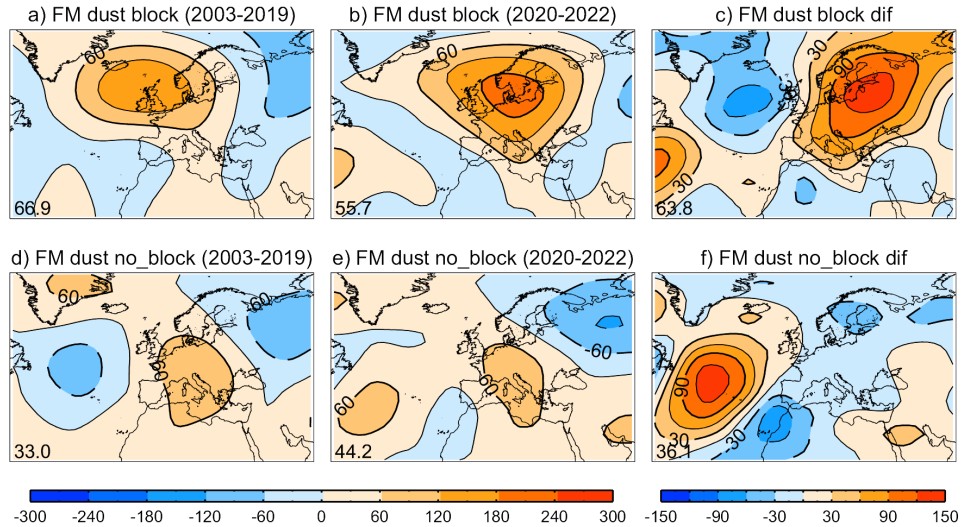

**Figure 7. Composites of Z500 anomalies (in m) for dust days of: (a), (d), and (c), the reference 2003-2019 period; (b), (e), and (h), the anomalous 2020-2022 period; (c), (f), and (i), the difference of composites between the 2020-2022 and 2003-2019 periods. Top and middle panels show the composites for dust days ((a), (b) and (c)) with and without blocking ((d), (e), and (f)) panels show the composites for dust days with and without blocking over the 25° W-30º E domain, respectively. Geopotential height fields have been obtained from NCEP/NCAR reanalysis.**

On the other hand, non-blocking dust days are favoured by high pressures over central Europe and the Mediterranean (Figures 7 (d), (e)). These situations denote high-pressure systems with no meridional reversals of the Z500 gradient, which is a distinctive feature of pure subtropical ridges (Sousa et al. 2021). This non-blocking pattern was recurrent during 2020-2022 and it actually concurred with more dust days than in the historical period, explaining the reduced intervention of blocking. The difference between the non-blocking dust days composites of Z500 for 2020-2022 and 2003-2019 indicates an anomalous occurrence and/or location of cut-off lows over the Canary Islands and eastern northern Africa during 2020-2022 (Figure 7 (f)). It is unclear whether subtropical ridges and cut-off lows represent independent phenomena (and hence their unusually high concurrence in 2020-2022 was by chance) or they can be dynamically linked, as in the case of blocking and cut-off lows. In any case, the results are consistent with those of Section 3.2.2.1, which report an increased percentage of dust events accompanied by cut-off lows during the anomalous period of 2020-2022. Therefore, while blocking was still the dominant



pattern associated with dust days in 2020-2022, an enhanced occurrence of subtropical ridges and cut-off lows at low latitudes also contributed to the outstanding frequency of dust intrusions.

## 4. Discussion and conclusions

In 2020-2022, several international agencies and media reported severe winter dust intrusions in Europe. Some of them
displayed a duration never recorded before and were referred to as "unusual", "historical", or "exceptional". Although winter dust intrusions are more common in the eastern Mediterranean, these events largely affected the western Mediterranean and Canary Islands. A record-breaking dust event occurred on 15-30 March 2022, with an AOD mean of 0.4. The event impacted on air quality, with large PM10 values (50-100 µg·m-3) at many air quality stations of western and central Europe, including those located on high mountains, such as the Pyrenees and the Alps. This event broke all records in terms of duration (14 days),
intensity, and affected area.

This work provides a catalogue of dust events over the western Mediterranean region and describes their characteristics during the climatological (2003-2019) and recent anomalous (2020-2022) periods. The criteria used to detect a dust event are those employed in previous studies of winter dust intrusions in the Mediterranean: the spatial average AOD must exceed 0.18 for at least three consecutive days. This study also contributes to the understanding of western Mediterranean dust events and their
synoptic and large-scale environments, which have been poorly addressed in the literature. The main results can be summarized as follows:

- Dust intrusions show a very irregular record and large differences between winter months. January is characterised by very low frequencies (only two events in 20 years). In February the number of dust events increased considerably, but almost half of them occurred in 2020-2022, denoting a high frequency of long calm
periods. In March, after an inactive period of four years (2016-2019), five dust events were recorded in 2020-2022, including two extreme ones. It is unclear if these intra-seasonal differences are meaningful or just reflect the effects of limited sampling (a short record with a low number of events).

- The percentage of western Mediterranean dust events accompanied by cut-off lows between the Canary Islands and the Iberian Peninsula is noticeable (more than half of all FM dust events), and substantially higher than that
reported for central and eastern Mediterranean. These results indicate that cut-off lows are a key synoptic driver for the occurrence of western Mediterranean dust intrusions in winter. During the anomalous 2020-2022 period, the cut-off low-dust association increased, and all March dust events of that period were concurrent with cut-off lows, suggesting an unusual frequency and/or efficiency of cut-off lows. In this sense, the cut-off lows of the 2020-2022 dust events were deeper (stronger anomalies) and extended to higher pressure levels than those of the
normal period. Consequently, the percentage of pure dust over the total available aerosol observations increased





in the lower troposphere (0-4 km) during recent dust events, which also reported a non-negligible percentage of pure dust in upper layers (4-6 km).

- Concerning the large-scale atmospheric circulation, western Mediterranean dust days are associated with enhanced Rex-like blocking activity over the mid latitudes of the Euro-Atlantic sector, being the probability of blocking occurrence more than double during dust days as compared to the climatology. This meridional pressure dipole favours the obstruction of the westerlies (which are shifted poleward by the blocking high) and the occurrence of cut-off lows at subtropical latitudes. Although European blocking is not a necessary condition for the occurrence of dust, it represents a favourable factor. Indeed, when the atmospheric circulation patterns of winter dust days are clustered, the leading group corresponds to the canonical pattern of European blocking.

- Enhanced mid-latitude European blocking activity and poleward jet configurations were favourable for the outstanding occurrence of western Mediterranean dust days during the 2020-2022 period. However, they cannot fully explain the anomalous frequency of winter dust days. A substantial fraction of dust days (particularly those in February 2020-2022) concurred with high-pressure structures at low latitudes, also called low-latitude blocks or subtropical ridges. Unlike mid-latitude European blocking, these high-pressure systems reinforce the zonal wind at their poleward flanks, favouring a mid-latitude intensification of the jet, rather than its poleward migration. Although subtropical ridges do not necessarily develop secluded troughs, our analyses suggest that they were frequently accompanied by cut-off lows. Further studies are encouraged to elucidate whether subtropical ridges and cut-off lows are dynamically connected, as in the case of blocking.

To assess whether the anomalous conditions of the 2020-2022 winters persisted during the winter of 2023 we have evaluated the available data for February and March 2023 (Figure S12 in Section S3 of the Supplement). The 2022-2023 winter was characterised by a severe drought in the north of Morocco, Algeria, and the Iberian Peninsula (JRC, 2023). A 6-day strong dust event occurred on 18-23 February 2023 with mean AOD of $0.31 \pm 0.09$, and a maximum daily AOD of 0.41 on 21 February (Figure S13 in Section S3 of the Supplement). Like for the dust events of the 2020-2022 period, dust uplift and its transport to the study region was associated with two concatenated cut-off lows in the vicinity of the Canary Islands. This dust event was also accompanied by positive Z500 anomalies over central Europe and the Mediterranean, like the patterns of low-latitude blocks and subtropical ridges. However, no dust event was recorded in March 2023, indicating that the unusually high frequency of western Mediterranean dust intrusions did not continue during the winter of 2023, which supports our choice of 2020-2022 as the anomalous period.

Winter dust events in the western Mediterranean constitute a good example of atmospheric processes interacting at very different scales: microscale to mesoscale processes, necessary for dust lifting and mobilization (Knippertz and Todd, 2012) from small dust sources located in Morocco and Algeria (Ginoux et al., 2012), synoptic-scale systems such as cut-off lows





driving its transport (Muñoz et al., 2020), and a large-scale environment favouring these synoptic disturbances and the persistence of high AOD values. The latter can occur under a variety of large-scale high-pressure systems, including European blocking with poleward shifted jets (as in the reference 2003-2019 period) or Mediterranean subtropical ridges with an intensified mid-latitude jet (e.g., as in the anomalous 2020-2022 period). Similarly, Cuevas et al. (2017) showed that Saharan
dust pulses towards the Mediterranean and the North Atlantic in summer were the result of the coupling between mesoscale dust uplift processes and longitudinal shifts of the Sahara heat low governed by the North African Dipole and Rossby waves.

This study that winter dust intrusions in the western Mediterranean are linked to meridional flows typically induced by subtropical cut-off lows and/or large-scale high-pressure systems at mid or low latitudes. The relationship between dust events and the synoptic-to-large scale atmospheric circulation is complex, like the dynamics of blocking, which involves non-linear
interactions between transient eddies and the large-scale flow (e.g., Kautz et al., 2022), as well as interactions with the ocean or the stratosphere, which are still not fully understood (e.g. Woollings et al., 2018, and references therein). Despite this, to the author's knowledge, this work represents the first comprehensive study showing a close relationship between Saharan dust events and different types of high-pressure systems, namely atmospheric blocking and subtropical ridges. Reliable forecasts of these weather systems would be useful for the mean-range predictability of dust events. We have shown that some of them
might lead to severe pollution episodes in long-range transported regions like Europe, stressing the added value of dust forecasts. The relationship between blocking, dust and air quality is also supported by recent studies describing atmospheric blocking influences on air stagnation and PM10 pollution in Europe (Garrido-Pérez et al., 2018). High ozone levels in Europe have also been related to blocking (Otero et al.,2022), as well as to the location of subtropical ridges (Ordónez et al. 2017). Therefore, blocks and ridges represent potential predictors of pollution events in Europe, as also shown by Maddison et al.
(2021) using statistical models.

The shocking images of European ski slopes covered by Saharan dust that were published by the media during the winters of 2021-2022 raised the question of whether these events are becoming more likely because of climate change. However, several reasons prevent making attribution statements at this stage. First, our catalogue of dust events is short and shows large temporal variations (such as the sharp increase of 2020-2022) and relatively long periods with no Saharan intrusions, which hamper the
detection of long-term trends. Secondly, the atmospheric weather systems linked to dust events do not exhibit robust trends and/or our confidence in their regional climate change responses is low. For example, blocking trends over the last decades are seasonally and spatially heterogeneous, with no robust evidence of generalized significant changes (Davini et al., 2012; Sousa et al., 2021) partially due to discrepancies among the detection indices (Barriopedro et al., 2010; Kautz et al. 2022, and references herein), but also to large internal variability on interannual-to-multidecadal time scales (e.g. Woollings et al., 2018),
which can make trends sensitive to the analysed period (Sousa et al., 2021). Although several studies point to future decreases in blocking frequency over the Northern Hemisphere, the spatial and seasonal patterns of change can be complex or depend on the blocking index and the characteristic addressed (frequency, duration, intensity, extension, etc.). For example, recent



studies have reported both significant decreases (Davini and D'Andrea 2020) and non-significant changes (Bacer et al. 2022) in the frequency and duration of Northern Hemisphere blocking during the 21st century, while other attributes such as blocking size may increase in response to climate change (Nabizadeh et al. 2019). On the other hand, subtropical ridges have been less addressed than blocking, and hence their observed and projected changes remain uncertain. Sousa et al. (2021) reported

regional increases in the winter frequency of subtropical ridges over the Euro-Mediterranean region, apparently associated with an eastward migration of low-latitude high-pressure systems towards the continent (Davini et al. 2012). However, these trends were not robust to the analysed period, likely suggesting a large influence of decadal variability. Finally, the detection of trends in cut-off lows is hampered by the diversity of detection approaches (which can be sensitive to methodological aspects such as the spatial resolution or the vertical level), the large interannual variability at regional scales and the influence of

internal modes of variability. For the European sector and the 1958-1998 period, no trends in the frequency of winter cut-of lows were reported by Nieto et al. (2007).

In addition to atmospheric dynamics, the sources and atmospheric loading of dust are sensitive to changes in other climate components and forcings, including vegetation and land use (IPCC, 2022). In this sense, the anomalous 2020-2022 period coincided with dry conditions and the lowest winter NDVI monthly values (Figure 1b). Warm Sea Surface Temperature (SST)

conditions, as those reported in the easternmost subtropical North Atlantic and the westernmost Mediterranean in the period 2003-2023 (Figure S14 in Section 3 of the Supplement) seem to play a crucial role in the development and evolution of Mediterranean cyclones, by fueling them with energy and water vapor (Stathopoulos et al., 2020), which may also favour dust intrusions (Flaounas et al. 2015). Finally, Thomas and Nigam (2018) reported a modest northward advance of the Sahara Desert during the boreal winter. Therefore, desertification processes and their complex interactions with the ocean, orography

and soil conditions over North Africa could also affect the frequency and/or severity of winter dust intrusions.

**Data availability**

The MODIS and MERRA-2 data used in this work can be freely obtained from https://giovanni,gsfc.nasa.gov/giovanni. CALIOP Level 2.5km Vertical Feature Mask (VFM) Version 3.1 and 4.2 product available from January to March 2007-2022 download from NASA's Atmospheric Science Data Center (ASDC, https://asdc.larc.nasa.gov/project/CALIPSO). NCEP-

NCAR Reanalysis data used in this work were obtained the NOAA PSL, Boulder, Colorado, USA, from their website at https://psl.noaa.gov/ (Kalnay et al., 1996). In the supplementary material is also considered: AOD direct-sun Version 3 Level 1.5 data can be download from the AERONET database (https://aeronet,gsfc,nasa.gov; Holben et al., 1998; Giles et al., 2019). Daily mean PM10 data used was obtained from the European Environment Agency's air quality database (AirBase; https://www.eea.europa.eu/data-and-maps/data/aqereporting-9). Red–green–blue (RGB) image animations from Meteosat

Second Generation (MSG) Spinning Enhanced Visible and InfraRed Imager (SEVIRI) from EUMESAT (and processed with the Man computer Interactive Data Access System, McIDAS) as well as dust forecasts from the MONARCH (Multiscale



Online Non-hydrostatic AtmospheRe Chemistry) are available through the WMO Barcelona Dust Regional Center (https://dust.aemet.es/products/daily-dust-products).

**Author contributions**

EC designed the study and coordinate the different contributions of the analysis. EC, DB and SB discussed the main results
and wrote the draft, with contributions all co-authors. RDG, JJB, OG and AB collected and prepared MODIS, MERRA-2 and AERONET datasets for the elaboration of the dust catalogue. SB collected and prepared the CALIOP datasets used in the dust catalogue. SAP and JJGA prepared the results on weather regimes. DB prepared and lead the discussion on meteorological drivers of dust events of the study. EW, DS and GGC collected and prepared the modelling and monitoring datasets included in the study cases considered in the Supplement. All co-authors reviewed and contributed to the final editing of the paper.

**Competing interests**

The authors declare that they have no conflict of interest.

**Acknowledgements**

The authors thank all the principal investigators and their staff, for establishing and maintaining the NASA and PHOTONS AERONET sites, the NASA MODIS and CALIOP mission, the NASA MERRA scientists and associated NASA personnel,
to produce the data used in this study. Also, EUMESAT and WMO are sincerely acknowledge for the provision of monitoring and forecasting products. Ongoing support for GrADS is provided by an omnibus grant jointly funded by the NSF, NOAA and NASA that forms the core support for all research at COLA. NCEP/NCAR data is provided by the NOAA/ESRL PSL, Boulder, Colorado, USA, from their Web site at https://psl.noaa.gov/. E. Cuevas, E. Werner, G. García-Castrillo, and S. Basart acknowledge the WMO Barcelona Dust Regional Center. S. Basart acknowledges CAMS-84 and CAMS2-82 (part of the
Copernicus Atmospheric Monitoring Services, CAMS).

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
