# Peer review of "Sharp increase of Saharan dust intrusions over the western Euro-Mediterranean in February-March 2020-2022 and associated atmospheric circulation"

_EGUsphere, 2023_

## Referee Comment (RC2)

**Review_report_egusphere_2023-1749**

The current study examines the meteorological drivers favored the occurrence of dust outbreaks in the western Mediterranean during winter periods over recent years (2020-2022). In winter, the occurrence of dust episodes is more common in the central/eastern Mediterranean in contrast to the western sector. The authors analyze/present a variety of reanalysis and observational datasets (observational, reanalysis) towards reaching to their goal. I have some concerns about the datasets which are utilized. Despite this, I believe that it is very interesting and constructive study, and it can be accepted for publication after revising the manuscript based on the following comments/suggestions:

1. **Page 6 – Line 12:** Why are you using the MODIS Collection 6 data and not those of 6.1?
2. **Page 6 (Line 24) – Page 7 (Line 5):** It would be useful here to elaborate how much your results are affected by "mixing" two different CALIPSO versions. I would remove Level 2.5km from the text because it is confusing (Level 2 – 5 km resolution along the satellite track).
3. **Section 2.2:** Can you explain why you are not using a more updated reanalysis dataset providing numerical products at finer spatial resolution (e.g., ERA5, GDAS)? I think that this is a very important issue since atmospheric patterns (not evident in the coarse NCEP/NCAR reanalysis dataset) can be revealed.
4. **Page 7 – Lines 18-19:** Can you rephrase this sentence? It is not so clear.
5. **Page 8 – Lines 11-12:** Do you mean the low-level jet or there is mistake in the pressure levels?
6. **Page 12 – Lines 1-16:** The authors state that they are processing the MODIS L3 AOD data. Which data are used exactly (daily or monthly)? Can you comment (show) how cloud contamination can "impact" your results considering that the analysis is representative for winter months? Have you checked the temporal availability of the MODIS data? I assume that due to extended cloud coverage there will be gaps throughout the study period. If so, this might have impact on the calculation of the mean and standard deviation values.
7. **Page 13 – Lines 4-17:** It would be useful to discuss further the maximum occurrences recorded in February 2016 and 2017. How much different was the atmospheric circulation in the aforementioned months? Are they other factors which can explain these maximum frequencies?
8. **Page 16 – Lines 13-14:** How much different are the atmospheric patterns presented here with those discussed in previous relevant studies?
9. **Page 16 – Lines 14-17:** I would propose to rephrase these sentences to be consistent with the relevant figures. What do you mean four concatenated cut-off lows? How are you excluding the possibility of a persistent low-pressure system? I would suggest discussing more the position and the strength of the anticyclones as well as the convergence zones.
10. **Page 17 – Lines 4-5:** Can you please rephrase this sentence?
11. **Figure 4:** It seems that between the clusters 1 and 2 many similarities in spatial terms exist and there are deviations on the relative frequencies. Nevertheless, this is not the case for the clusters 3 and 4, as already stated in the manuscript. Can you please interpret the observed inconsistencies?
12. **Figure 5 and the relevant discussion:** The authors state that "*…that some WRs do not have a direct apparent correspondence with the clusters of Figure 4 (e.g., GL, ZO).*". I am confused with this part of the study. How much can affect this inconsistency the connection between the patterns that you have obtained from the cluster analysis and the weather regimes of Grams et al. (2017)? If I am not missing something, in the latter study it is not considered the dust transport from N. Africa towards the region of interest.
13. **Page 20 – Lines 1-4:** It would be easy to reproduce the maps with winds at 10 meters in order to check in which regions the wind speeds exceed the thresholds.
14. **Page 20 – Lines 6-7:** I would remove or rephrase the '*…before being absorbed by the general circulation.*'
15. **Section 4:** Please consider splitting this section in "Discussion" and "Conclusions". Also, I believe that the part of the text after the bullets can be reduced by summarizing the main findings and outcomes.

---

## Author Comment (AC1)

**Reply to Diana Francis**

1. In a previous work we have identified atmospheric rivers as a main driver for this kind of Saharan dust intrusions. Would be good to see a discussion in this paper on this and how the identified atmospheric circulation is different/similar to it: https://www.sciencedirect.com/science/article/pii/S0169809521005159

Thanks, Dr Diana Francis, for the suggestion. The two intense episodes of Saharan dust transport during February 2021 were associated with Atmospheric Rivers (ARs), which were proposed as a potential driver of (moist) dust events in Europe by Francis et al. (2022). This study has been cited in the revised version of the manuscript. We have also included a short paragraph in Section 3 of the revised manuscript summarising the following points:

1. According to Francis et al. (2022), the majority (~78%) of AR days coincided with strong to extreme dust events, but only ~18% of the dust episodes co-occurred with AR events. For the revised version of the manuscript, we have re-assessed this relationship by using our catalogue of February-March dust events over the western Euro-Mediterranean and ARs, identified visually as long and narrow structures with vertically integrated water vapour higher than 20 Kg m$^{-2}$ (Gimeno et al., 2014). The results confirm that less than one-third of the dust events of 2003-2022 concurred with ARs (see two examples in Figure R1.1). This linkage is even weaker when dust days are considered (~17% of the dust days with an AR in 2020-2022). This probability is like that reported by Francis et al. (2022) and lower than that obtained by conditioning on blocking (~50%), suggesting that the latter exerts a more significant control. Despite this, we agree that ARs might play an important role in the wet deposition of dust, as was the case during some of the extreme events analysed in the manuscript (and in Francis et al., 2022). Therefore, in the revised manuscript, we encourage additional studies to address the similarities and differences between dry and wet dust events.

2. We believe that ARs are fully independent of the atmospheric circulation drivers considered in our manuscript (cut-off lows and blocking). Indeed, the synoptic analyses

of the two case studies described by Francis et al. (2022) show a wavy jet with either a ridge over central Mediterranean or an omega block over central Europe, both accompanied by troughs over its western and/or eastern flanks, which typically evolve into cut-off lows. This high-low pressure configuration is consistent with that reported in the submitted manuscript (see Figure 6) and with the underlying drivers considered therein (blocking and cut-off lows). It also resembles the composites for Mediterranean ARs described in Lorente-Plazas et al. (2019). The reviewed literature suggests that northwestern African cut-off lows are modulated by the large-scale flow blocking (Nieto et al., 2007), with the southerly flow in between promoting ARs (Lorente-Plazas et al., 2019) and the advection of Saharan dust (Francis et al., 2022). The influence of blocking on ARs. For example, Benedict et al. (2019) found that North Pacific blocking modulates AR probabilities along the North American west coast. The effect of the large-scale flow configurations on AR occurrence and landfall has also been addressed in other studies. For example, Pasquier et al. (2018) found that during blocking-like weather regimes (WRs), such as Scandinavian blocking, the probability of AR landfall increases over the western Mediterranean and northern Africa. In contrast, cyclonic zonal flow (ZO) favours AR landfalls at comparatively higher latitudes, which is consistent with Figure 5 of the submitted manuscript. From this perspective, ARs could be considered a consequence (or a coupled element) of the enhanced meridional flow instigated by the high-low pressure dipole. This has been briefly stated in the revised text.

[Figure]

*Figure R1.1: Examples of daily snapshots ARs diagnosed from vertically integrated water vapour from the NCEP/NCAR reanalysis: 30 March 2022 (left) and 21 February 2004 (right).*

**References**

Benedict, J. J., Clement, A. C. and Medeiros, B.: Atmospheric blocking and other large-scale precursor patterns of landfalling atmospheric rivers in the North Pacific: A CESM2 study. Journal of Geophysical Research: Atmospheres, 124, https://doi.org/10.1029/2019JD030790, 2019.

Francis, D., Fonseca, R., Nelli, N., Bozkurt, D., Picard, G., Guan, B.: Atmospheric rivers drive exceptional Saharan dust transport towards Europe, Atmospheric Research, 266, 105959, https://doi.org/10.1016/j.atmosres.2021.105959, 2022.

Gimeno, L., Nieto, R., Vázquez, M. and Lavers, D.: Atmospheric rivers: a mini-review. Frontiers in Earth Science, 2, https://doi.org/10.3389/feart.2014.00002, 2014

Lorente-Plazas, R., Montavez, J. P., Ramos, A. M., Jerez, S., Trigo, R. M. and Jimenez-Guerrero, P.: Unusual atmospheric-river-like structures coming from Africa induce extreme precipitation over the western Mediterranean Sea. Journal of Geophysical Research: Atmospheres, 125, e2019JD031280, https://doi.org/10.1029/2019JD031280, 2020.

Nieto, R., L. Gimeno, L. de la Torre, Ribera, P., Barriopedro, D., García-Herrera, R., Serrano, A., Gordillo, A., Redaño, A. and Lorente, J.: Interannual variability of cut-off low systems over the European sector: The role of blocking and the Northern Hemisphere circulation modes, Meteorol. Atmos. Phys., 96, 85–101, https://doi.org/10.1007/s00703-006-0222-7, 2007.

Pasquier, J. T., Pfahl, S. and Grams, C. M.: Modulation of atmospheric river occurrence and associated precipitation extremes in the North Atlantic Region by European weather regimes. Geophysical Research Letters, 46, 1014–1023, https://doi.org/10.1029/2018GL081194, 2019.

---

## Author Comment (AC2)

We are grateful for the positive evaluation and constructive comments, which have helped improve the manuscript. Our replies are shown below in blue, after the Reviewer's comments (in black).

**Reply to Reviewer #2**

1. The current study examines the meteorological drivers favored the occurrence of dust outbreaks in the western Mediterranean during winter periods over recent years (2020-2022). In winter, the occurrence of dust episodes is more common in the central/eastern Mediterranean in contrast to the western sector. The authors analyze/present a variety of reanalysis and observational datasets (observational, reanalysis) towards reaching to their goal. I have some concerns about the datasets which are utilized. Despite this, I believe that it is very interesting and constructive study, and it can be accepted for publication after revising the manuscript based on the following comments/suggestions:

Please, be aware that following the recommendation of one of the reviewers of the present manuscript the title is modified as *"Sharp increase of Saharan dust intrusions over the western Euro-Mediterranean in February-March 2020-2022 and associated atmospheric circulation".*

2. Page 6 – Line 12: Why are you using the MODIS Collection 6 data and not those of 6.1?

Thanks for noting the typo. We have used the MYD08_D3 v6.1 product, i.e. the level-3 MODIS gridded atmosphere daily global joint product (Collection 6.1). This has been corrected in the revised manuscript, including the following information in Section 2.1:

*"We have used the NASA MODIS/Aqua daily global aerosol product (Collection 6.1), specifically, the AOD (at 550 nm) Combined for Land and Ocean product (Sayer et al., 2013), available since 2003 at 1º x 1º horizontal resolution."*

3. Page 6 (Line 24) – Page 7 (Line 5): It would be useful here to elaborate how much your results are affected by "mixing" two different CALIPSO versions. I would remove Level

2.5km from the text because it is confusing (Level 2 – 5 km resolution along the satellite track).

Data generation and distribution of the Cloud-Aerosol Lidar and Infrared Pathfinder Satellite Observations (CALIPSO) Lidar Level 2 Vertical Feature Mask (VFM) is done through the NASA Atmospheric Science Data (ASC; https://asdc.larc.nasa.gov/project/CALIPSO). CALIPSO Level 2 VFM products has associated different versions. By the time in which the results of the manuscript were prepared, the only dataset covering most of the period was the CALIPSO VFM v3.x (2007-2021) and the 2022 was only available in v4.2. As it is shown in the comparison between the two CALIPSO versions (https://www-calipso.larc.nasa.gov/resources/calipso_users_guide/qs/cal_lid_l2_all_v4-20.php), the dust aerosol typing remains consistent. Noted that only recently (in December 2023), NASA ASC published a new CALIPSO Level VFM v4.5 (i.e. CAL_LID_L2_VFM-Standard-V4-51_V4-51) that is covering 2007-2022. Unfortunately, because the short notice, we are not in time to include this dataset in the revised manuscript. However, no substantial changes are considered in this v4.5 with respect to v3.x or v4.21 for the dust typing (see https://www-calipso.larc.nasa.gov/resources/calipso_users_guide/qs/cal_lid_l2_all_v4-51_qs.php). Then, we expect comparable results. In fact, a visual inspection of the results of the three CALIPSO Level 2 VFM versions for the dust episodes occurred in 2021 (http://www-calipso.larc.nasa.gov/about/) confirm it.

Following the Reviewer #2's suggestion, the description of the CALIOP products is revised as follows in Section 2.1 of the revised manuscript:

*"Here, we use the available CALIPSO Lidar Level 2 VFM product from NASA Atmospheric Science Data Center, which includes Version 4.2 (2007-2021) and Version 3.1 (for year 2022). Please, note that these are the available CALIPSO datasets in the NASA Atmospheric Science Data (https://asdc.larc.nasa.gov/project/CALIPSO, last access 15 June 2023) by the time the results were processed. Despite the use of two different processing algorithms, the comparison between the two versions shows dust typing remains consistent (https://www-calipso.larc.nasa.gov/resources/calipso_users_guide/qs/cal_lid_l2_all_v4-20.php, last access 15 September 2023)."*

4. Section 2.2: Can you explain why you are not using a more updated reanalysis dataset providing numerical products at finer spatial resolution (e.g., ERA5, GDAS)? I think that this is a very important issue since atmospheric patterns (not evident in the coarse NCEP/NCAR reanalysis dataset) can be revealed.

In this study we deal with synoptic and large-scale atmospheric circulation patterns. At these spatial scales global reanalyses provide similar patterns over the Euro-Atlantic sector. This is also true for the weather systems addressed in the manuscript, including blocking (see an inter-reanalysis comparison in Woollings et al., 2018) and the jet stream (see e.g. Barriopedro et al., 2023). We have repeated some of the analysis with ERA5 (see for example Figure R2.1), obtaining almost identical results. This has been stressed in the revised manuscript:

*"the results of the atmospheric circulation analyses are robust to the study period (e.g. the 2003-2022 dust period) and the atmospheric reanalysis employed (e.g. ERA5; Hersbach et al., 2020)"*.

[Figure]

*Figure R2.1. As Figure 6 of the main text but for ERA5.*

5. Page 7 – Lines 18-19: Can you rephrase this sentence? It is not so clear.

The sentence has been revised in Section 3.1 of the revised manuscript as follows:

*"A day is assigned to a given WR if the respective index is greater than 1 and higher than that of all other WRs".*

6. Page 8 – Lines 11-12: Do you mean the low-level jet or there is mistake in the pressure levels?

The eddy-driven jet stream has been diagnosed with zonal wind data averaged between 925 and 700 hPa. This is a standard procedure and hence we have not modified the text. Zonal wind averages over the low-troposphere are often employed to emphasise the eddy-driven jet (which has a barotropic structure through the troposphere) and avoid the detection of the subtropical jet, which peaks in intensity in the upper troposphere (see e.g. Woollings et al. 2010 and references therein). Differently, the diagnosis of low-level jets requires finer information about the two wind components and the vertical structure of the wind in the lower troposphere (e.g. Bonner, 1986). Note that low-level jets, if present, would be filtered out in our approach by the spatio-temporal averages of the zonal wind.

7. Page 12 – Lines 1-16: The authors state that they are processing the MODIS L3 AOD data. Which data are used exactly (daily or monthly)? Can you comment (show) how cloud contamination can "impact" your results considering that the analysis is representative for winter months? Have you checked the temporal availability of the MODIS data? I assume that due to extended cloud coverage there will be gaps throughout the study period. If so, this might have impact on the calculation of the mean and standard deviation values.

For the analysis, it is considered the NASA MODIS/Aqua global daily AOD at 550nm Combined Land and Ocean product (Sayer et al., 2013), available since 2003 at 1º x 1º horizontal resolution (i.e. MYD08_D3). This is detailed in the revised manuscript. It is well-known that wintertime is the period of maximum rainfall and cloud cover in the region, and this affects remote sensing retrievals (e.g. Basart et al., 2009; Gkikas et al., 2016). Because MODIS can provide limited coverage during wintertime in the study region because of the presence of clouds (see Gkikas et al., 2016; Basart et al., 2009), MERRA-2 reanalysis dust product (which provides representative and complete dust fields in space and time) is used to support the results obtained with MODIS about the anomaly of events identified in 2020-2022 with respect 2003-2019. This is described in Section 2 of the revised manuscript:

*"the identification and characterisation of dust events (Section 2.1) relies on satellite-based MODIS aerosol products over the available period of observations (2003-2022). Because of the limitations associated with satellite-based products to capture the daily cycle (i.e. satellite overpasses or cloud contamination), the analysis of dust events has also been done with the MERRA-2 global reanalysis (Randles et al., 2017), confirming the exceptional dust activity of the 2020-2022 period obtained from MODIS (Section S1 of the Supplement)."*

8. Page 13 – Lines 4-17: It would be useful to discuss further the maximum occurrences recorded in February 2016 and 2017. How much different was the atmospheric circulation in the aforementioned months? Are they other factors which can explain these maximum frequencies?

Precisely the dust event of February 20-24, 2017 is analysed in detail in Section S1 of the Supplement as the first case study of the three dust events labelled as extreme. These results are shown in S1 of the submitted version of the Supplement.

Below (Figure R2.2) we include information on the atmospheric circulation corresponding to the dust event of February 29-24, 2017, which is compared with that of February 21-23, 2016.

[Figure]

*Figure R2.2 NCEP reanalysis mean Z500 for each day or subperiods of days for the two following dust events: February, 20-24, 2017 ((a), (b), (c) and (d)) and February, 21-23, 2016 ((e), (f), and (g)).*

[Figure]

*Figure R2.3. NCEP-Reanalysis geopotential height (Z) anomalies (m) at 925 hPa ((a) and (e), 500 hPa ((b), and (f)), 200 hPa ((c), and (g)), and 100 hPa ((d), and (h)), with respect to the reference period 1991-2020, for the following dust events occurred in the western Euro-Mediterranean. February 20-24, 2017; and February 21-23, 2016. Note that Z anomalies at each level, for brevity, have been averaged over the duration of each dust event (at least 3 three days).*

Like the other case analyses, the February 21-23 ,2016 event shows a cut-off low to the west of the Atlas Mountains with very weak signal at 925hPa, and a downstream subtropical ridge, confirming the large-scale atmospheric patterns found in this study

9.  Page 16 – Lines 13-14: How much different are the atmospheric patterns presented here with those discussed in previous relevant studies?

As far as we know, previous studies have not addressed in a systematic way the synoptic patterns (in particular, cut-off lows) associated with dust intrusions over western Mediterranean in February-March. We stated in the main text that: *"Winter dust intrusions over the western Mediterranean have received little attention and existing studies on the associated atmospheric circulation have mainly focused on synoptic systems during individual case studies."*

The two articles referenced in the text (Fernández et al., 2019; Oduber et al., 2019) did not describe the atmospheric process causing the dust events, since they focused on the optical characterization of dust and its impact in non-African regions. On the other hand, Francis et al. (2022) attributed the dust event included in our study (March 15-30, 2022) to the presence of a cut-off low located between the Canary Islands and the Iberian Peninsula.

As the Referee #2's statement refers to three dust events only, in Section 3 of the revised manuscript we have clarified that the results of these case studies should not generalised: *"Although these results should not be generalised to all dust events, the inspection of case studies suggests that cut-off lows and downstream high-pressure systems are commonly involved during WEM dust events."*

10. Page 16 – Lines 14-17: I would propose to rephrase these sentences to be consistent with the relevant figures. What do you mean four concatenated cut-off lows? How are you excluding the possibility of a persistent low-pressure system? I would suggest discussing more the position and the strength of the anticyclones as well as the convergence zones.

The identification of cut-off lows was carried out by visual inspection, following the conceptual model of Nieto et al. (2005). As stated in the text, these structures reach their maximum expression in the upper troposphere and the distinctive upper-level low does not often reach

the surface. The absence of a surface low is the main difference between cut-off lows and extratropical cyclones. By "concatenated" cut-off lows we mean a clustering of cut-off lows (i.e. a sequence of multiple cut-off lows following similar paths). For each dust event, the identification of cut-off lows was determined by using 6-hourly fields of geopotential height and animated 15-min EUMETSAT RGB dust images. This can be seen in the three timelapses corresponding to the three extreme dust events that have been our three case studies (see the associated timelapses videos at https://repositorio.aemet.es/handle/20.500.11765/15054).

According to the Reviewer #2's suggestion, we have rephrased the corresponding sentences in the revised manuscript:

*"In most cases, cut-off lows moved eastward from the subtropical North Atlantic to the western Mediterranean (see, e.g. the cut-off low to the west of the Atlas Mountains during the events of 20-24 February 2017 and 27-31 March 2021; Section S1 of the Supplement). The same event can comprise two or more successive cut-off lows. This situation was identified in at least one-third of the 2003-2019 dust events, but in almost all March dust events of the 2020-2022 period (see the sequence of cut-off lows over the coasts of Morocco and Algeria in the 15-31 March 2022 event; Section S1 of the Supplement)".*

11. Page 17 – Lines 4-5: Can you please rephrase this sentence?

Thanks for noting. The sentence has been rephrased in the revised manuscript as follows:

*"Figure 4 shows the two most recurrent Z500 anomaly patterns obtained from the k-means clustering of FM dust days for the 2003-2019 and 2020-2022 periods".*

12. Figure 4: It seems that between the clusters 1 and 2 many similarities in spatial terms exist and there are deviations on the relative frequencies. Nevertheless, this is not the case for the clusters 3 and 4, as already stated in the manuscript. Can you please interpret the observed inconsistencies?

Figure 4 and the associated text have been modified in the revised manuscript following the recommendations of Reviewer #2. We have only defined two clusters, which yields a robust partitioning of the data while still retaining the major features of the dominant patterns associated with dust events (see Figure R2.2). This has been stated in Sections 2.2 and 3.2. of the revised manuscript:

*"The selected number of clusters was limited to two, considering the relatively low number of dust days, particularly for the recent period of 2020-2022. The method assigns each dust day to*

*one of the two clusters, allowing us to explore the two main Z500 patterns associated with WEM dust intrusions"*

*[...]"we have only retained two clusters, which yields a robust partitioning of the data while still retaining the major features of the dominant patterns associated with dust events"*.

New cluster #1 shows similar spatial patterns in the two periods, and is more frequent in the anomalous period, whereas the spatial patterns of cluster #2 differ between the two periods. This suggest that blocking (cluster #1) is a recurrent driver of dust intrusions, being present in both the reference and anomalous period. Differences between clusters #2 suggest that other favourable atmospheric patterns (e.g. cut-off lows and subtropical ridges) occurred in the anomalous period, and with higher frequency than in the reference period. Therefore, the enhanced dust activity of the recent 2020-2022 period can partially be explained by a high frequency of recurrent (blocking) and less common (subtropical ridges) favourable patterns. The former (latter) was particularly prominent in March (February) 2020-2022.

[Figure]

*Figure R2.2 (new Figure 4). Two clusters (C#1 and C#2) of Z500 anomalies (in m) for the FM dust days of the normal-dust 2003-2019 (a-b) and the anomalous-dust 2020-2022 (c-d) period. The relative frequency of each cluster (in % with respect to the total number of dust days of each period) is indicated on the top.*

13. Figure 5 and the relevant discussion: The authors state that "…that some WRs do not have a direct apparent correspondence with the clusters of Figure 4 (e.g., GL, ZO).". I am confused with this part of the study. How much can affect this inconsistency the connection between the patterns that you have obtained from the cluster analysis and the weather regimes of Grams et al. (2017)? If I am not missing something, in the latter study it is not considered the dust transport from N. Africa towards the region of interest.

These two analyses are complementary. The cluster analysis (Figure 4) addresses the atmospheric patterns responsible for dust intrusions, allowing us to discriminate multiple flow configurations leading to dust events. In this procedure, the focus is on dust events since we only consider dust days. The so-identified configurations do not necessarily match with recurrent synoptic patterns responsible for the day-to-day variability of the extratropical circulation. This assessment is provided by the weather regime (WR) analysis (Figure 5), which uses all days of the analysed period for the definition of WR. WRs allow us to infer if dust can occur preferentially under specific large-scale preferred patterns. This has been clarified in Section 2.2 of the revised manuscript as follows:

*"First, we have carried out a characterisation of the synoptic patterns associated with dust events in order to discriminate different flow configurations leading to dust events"* (…) *"To categorise the large-scale atmospheric circulation in a limited number of recurrent weather regimes (WRs) we have followed Grams et al. (2017), which uses an extended year-round classification of the Euro-Atlantic atmospheric circulation in seven WRs"*

One should not expect a perfect correspondence between the drivers (clusters) and favourable recurrent patterns (WRs) because 1) dust events are relatively uncommon (much less common than WRs), and 2) our clustering analysis indicates that there are several large-scale drivers of dust. As stated by the reviewer, the definition of WRs does not consider dust events only (as the clustering does). Despite this, we find a connection between both approaches. In particular, dust events tend to occur during WRs featuring high-pressure systems at different latitudes (i.e. European Blocking, EuBl; and Atlantic Ridge, AR), which is confirmed by the clustering analysis (cluster #2). The ZO pattern is also consistent with some circulation features of cluster #1, particularly one of the reference periods (Figure 4a). Therefore, the two approaches are consistent, which adds robustness to the analysis. We have revised the text Section 3.2 accordingly in the revised manuscript, including the following:

*"The dust-related configurations identified in Figure 4 do not necessarily match with the dominant large-scale patterns responsible for the day-to-day variability of the extratropical circulation (WRs)"*

*"We note that none of the WRs coincide exactly with the dust-related patterns of Figure 4, which is reflected in the dispersion of dust days across different WRs. As such, dust intrusions can occur under different WRs, stressing again the multiplicity of large-scale patterns compatible with dust events"*

*"Therefore, dust intrusions can occur during WRs featuring high-pressure systems at different latitudes (mainly European Blocking, EuBl, and Atlantic Ridge, AR), which is consistent with the clustering analysis".*

14. Page 20 – Lines 1-4: It would be easy to reproduce the maps with winds at 10 meters in order to check in which regions the wind speeds exceed the thresholds.

We agree with the Referee #2. Figure R2.3 is included in the Supplement and referred in Section 3 of the revised manuscript.

[Figure]

*Figure R2.3 (Figure S15 added in the Supplement of the revised manuscript). NCEP Reanalysis vector wind for March 27, 2021: (a) 00 UTC; (b) 06 UTC; (c) 12 UTC; and (d) 18 UTC; Wind speed*

*scale ranges from 5 to 12.5 m·s⁻¹ that is the range of wind speed threshold for dust mobilization in the west Sahara, which is between 5 and 12.5 m·s⁻¹ (Helgren and Prospero, 1987).*

*In the revised manuscript, we have added the following text after the sentence "…they are intense enough to generate surface winds exceeding the wind speed threshold for dust mobilization in western Sahara, which is between 5 and 12.5 m·s-1 (Helgren and Prospero, 1987)":*

*"This is supported by 6-hourly surface wind data from NCEP/NCAR reanalysis over Morocco, Algeria, Western Sahara and Mauritania (see Figure S15 of the Supplement)".*

*In addition, we have included, at the end of Section 2.1 of the revised manuscript the following lines:*

*"The identification of dust hotspots was performed using the EUMETSAT RGB dust product (Met Office; EUMETSAT, 2022). This dataset contains RGB dust images from Meteosat Second Generation satellites over the full disc at a frequency of 15 minutes".*

*In Section 3.2.2 of the revised manuscript is also added the following lines:*

*"In the three case studies analysed, the geographical location as well as the activation time of each dust hotspot has also been identified manually (Schepanski et al. 2007, 2009, 2012) by using the 15-min EUMETSAT RGB dust animations (see timelapse videos at https://repositorio.aemet.es/handle/20.500.11765/15054)."*

15. Page 20 – Lines 6-7: I would remove or rephrase the '…before being absorbed by the general circulation.'

*Amended in the revised manuscript following the Referee #2's suggestion.*

16. Section 4: Please consider splitting this section in "Discussion" and "Conclusions". Also, I believe that the part of the text after the bullets can be reduced by summarizing the main findings and outcomes.

*Following the Referee #2's suggestion, the revised manuscript considers two sections. Also, the length of the text has been shortened, as suggested.*

**References**

*Basart, S., Pérez, C., Cuevas, E., Baldasano, J. M., and Gobbi, G. P.: Aerosol characterization in Northern Africa, Northeastern Atlantic, Mediterranean Basin and Middle East from direct-sun*

*AERONET observations, Atmos. Chem. Phys., 9, 8265–8282, https://doi.org/10.5194/acp-9-8265-2009, 2009.*

*Bonner, W.D. (1986). Climatology of the low level jet. Mon. Weather Rev., 833-850*

*Met Office; EUMETSAT (2022): MSG: Dust imagery in the RGB channels over the full disc at 41.5 degrees East (LEDF41, upto 0900 UTC 1st June 2022). NERC EDS Centre for Environmental Data Analysis,                    date                    of                    citation. https://catalogue.ceda.ac.uk/uuid/b1dacc09b42f4d8ab492c5d5c751efa9.*

*Woollings, T., Hannachi, A., Hoskins, B. (2010). The variability of the North Atlantic eddy-driven jet stream. Q. J. R. Meteorol. Soc., 136, 856–868*

---

## Author Comment (AC3)

We are grateful to the referee for the review, which has contributed to improve the manuscript. Below we provide a point-by-point answer (in blue) to the reviewer's comments (in black).

**Reply to Reviewer #1**

Here, the role that the atmospheric circulation probably played during extreme dust intrusions in the western Mediterranean is analysed. This work proposes interesting ideas regarding the sources of such rare dust events, however, in my opinion, there are significant problems that need to be addressed. Thus, I would recommend a major revision of the manuscript according to the following comments, which I hope the authors will find useful.

**Major Comments:**

1.  Firstly, the selection of the datasets and the time periods present inconsistencies: For the dust events, MERRA-2 dataset was used, while for the atmospheric circulation NCEP/NCAR was used. These two different reanalyses may exhibit such differences in their internal variability that could make the dust transport and geopotential height fields incomparable. Could you please justify why we didn't use datasets from the same reanalysis, also considering the higher resolution of MERRA-2?

We would like to highlight that MODIS aerosol product (i.e. aerosol optical depth, AOD, at 550nm) is the reference dataset used to identify the dust events/days. Considering that MODIS can provide limited coverage during wintertime because of the presence of clouds, MERRA-2 reanalysis (which provides representative and complete dust fields in space and time) is used to support the results obtained with MODIS about the anomaly of the dust events identified in 2020-2022 with respect 2003-2019.

The main objective of the present study is to investigate about the atmospheric circulation patterns associated (I.e. geopotential heights) to the previous identified "dusty" days. Overall,

the current available atmospheric reanalysis shows comparable results in terms of synoptic and global atmospheric patterns (see: https://climatedataguide.ucar.edu/climate-data/atmospheric-reanalysis-overview-comparison-tables#Table, Last connection 13 December 2023). NCEP/NCAR reanalysis is a well-established and operational product used in many publications focusing on the analysis of atmospheric circulation patterns in the study region considered (see Cuevas et al., 2017).

To avoid any confusion or misunderstanding to the reader, the revised manuscript considers only MODIS, and the results of MERRA-2 are moved to the Supplement. The main text and associated figures have been revised including a reference to the consistency of the results between MODIS and MERRA-2 identifying the dust events. To this end, Figures 1 and 2 (shown below as Figures R1.1 and R1.2) have been recomputed with MODIS AOD data in the revised manuscript, thus removing the original results and discussion considering MERRA-2. The main text is revised including the following text in Section 3.1:

*"The identification and characterisation of dust events (Section 2.1) relies on satellite-based MODIS/Aqua aerosol products over the available period of observations (2003-2022). Because of the coverage limitations associated to satellite-based products to capture the daily cycle (i.e. satellite overpasses or cloud contamination), the same dust event analysis has also been done with the MERRA-2 global reanalysis (Randles et al., 2017), confirming the exceptional dust activity of the 2020-2022 period obtained from MODIS (Section S1 of the Supplement)."*

[Figure]

*Figure R1.1 (Figure 1 in the revised manuscript). MODIS/Aqua AOD at 550 nm for: (a, e) December, (b, f) January, (c, g) February, and (d, h) March of the 2002-2019 period (a-d) and the 2020-2022 period (e-h). The black box delimits the spatial domain [35°-50°N, 20°W-5°E].*

[Figure]

*Figure R1.2 (Figure 2 in the revised manuscript). Time series of MODIS/Aqua AOD at 550 nm for each month of the extended winters (i.e., December, January, February, and March) of the 2003-2022 period and for the western Euro-Atlantic [35°-50°N, 20°W-5°E].*

To further clarify that dust events are diagnosed from MODIS, we have introduced some changes in Section 2.1 in the revised manuscript as follows:

*"We have used the NASA MODIS/Aqua daily global aerosol joint product (Collection 6.1), specifically the Combined AOD (at 550 nm) for Land and Ocean product (Sayer et al., 2013), available since 2003 at 1º x 1º horizontal resolution."*

*"For the assessment of the meridional transport of dust, monthly mean AOD values have been computed for each month of the extended winters (from December to March) of 2003-2022 and over a large domain [27-60ºN, 30°W-36°E], which encompasses northern Africa, the Mediterranean basin, Europe and the eastern North Atlantic. The identification of dust events in WEM, defined as [35-50°N, 20°W-5°E], is carried out by using daily MODIS AOD at 550 nm for the available period 2003-2022."*

2. In addition, the analysis is limited to 2003-2022, while data are available since 1980. Please justify why you limited your analysis to such a short period.

Until 2002, MERRA-2 assimilated bias-corrected AOD derived from the 25-yr record of AVHRR radiances (Heidinger et al. 2014), which only provided AOD retrievals over the oceans for 1997-2010. Other assimilated products include: 1) AERONET AOD Version 3 direct-sun quality-assured (i.e. level 2) (Holben et al. 1998) since 1999; 2) bias-corrected AOD derived from MODIS level-2 radiances available since 2000, first from the Terra spacecraft only, and after 2002 also from the Aqua spacecraft. Since 2007, CALIPSO aerosol profiles are assimilated as well. Accordingly, the temporal changes in the number and type of AOD observations assimilated in MERRA-2 do not

guarantee the consistency and homogeneity of the historical aerosol/dust series derived from this reanalysis product.

3. Finally, the selection of "winter" in the title and in the text is quite confusing, as in some parts of the text DJFM and/or JFM and/or FM are analysed. Please reconsider a more uniform and self-explanatory presentation of the results. In addition, I would suggest removing "winter" from the title or replace it with a more adequate phrase.

The authors agree with the referee. Both the title and the text of the manuscript have been modified to make it clear that this study focuses on the months of February and March.

The new title of the paper is *"Sharp increase of Saharan dust intrusions over the western Euro-Mediterranean in February-March 2020-2022 and associated atmospheric circulation".*

In the revised manuscript, the choice of these two winter months has been justified at the beginning of the results section after analysing all months of the extended winter (from December to March). From this analysis we find that dust intrusions over the western Mediterranean are small and stable (low interannual variability) in December and January. More importantly, there are no substantial changes in activity during 2020-2022, compared to 2003-2019. This has been clarified in Section 2.1 of the revised manuscript: the revised:

*"As the frequency of dust intrusions in WEM is almost negligible in December and January (Figures 1 (a-b), (e-f)), from now on the analyses will focus on the months of February and March only. Our choice of the wintertime, defined as February-March (FM), is supported by both climatological arguments (higher AOD in late than in early winter; Figure 2) and the degree of extremeness of the 2020-2022 period as compared to the 2003-2019 period (see Section S1 of the Supplement). Additional analyses based on MERRA-2 (Figure S11) confirm that FM 2020-2022 over WEM was unprecedented since at least 1980."*

The full manuscript is revised to be coherent with the February-March wintertime definition in terms of both terminology and analysis.

**Specific Comments**

1. Page 4, line 8: The leeward side of the Atlas Mountains is a favourable area of cyclogenesis. I do not agree with this statement. Please clarify.

The sentence mentioned by the reviewer (line 8, page 4) reads as follows: *"Indeed, dust emission in West Sahara (the origin of most of the dust events reaching the western Mediterranean, e.g.*

*Gkikas et al., 2016) occurs far away from well-known cyclogenesis regions, suggesting that dust transport might be rather associated with deep North Atlantic cyclones (Flaounas et al., 2022) or upper-level cut-off lows in the western Mediterranean (Portmann 10 et al., 2021)".* In these lines, it's mentioned that cyclogenesis occurs far away of West Sahara, as you mentioned.

Atlas Mountains are also mentioned in the following sentences of the original manuscript:

*1. "Cyclones on the leeward side of the Atlas Mountains, also known as "Sharav cyclones" (Winstanley, 1972), and more recently as North African cyclones or Saharan cyclones are also associated with dust storms in winter (Flaounas et al., 2022; and references herein)".*

*2. "Karam et al. (2009) described a strong cyclogenesis over the southern side of the Atlas Mountains associated with the African dust intrusion in February 2007 over western Mediterranean".*

*3. "In the first two cases (20-24 February 2017, and 27-31 March 2021) we identified a single cut-off low to the west of the Atlas Mountains, whereas in the third case (15-31 March 2022) four concatenated cut-off lows moved eastwards from the Atlantic to the Mediterranean coasts of Morocco and Algeria".*

In none of these sentences we state that the Atlas Mountains is a region of cyclogenesis.

To avoid any misunderstanding to the reader, in the revised manuscript the sentence is being modified as follows

*"However, the release of dust from western Sahara, which is the primary source of dust events reaching the western Mediterranean (e.g. Gkikas et al., 2016), takes place at a considerable distance from recognized cyclogenesis areas. This implies that the transport of dust may be more closely linked to intense North Atlantic cyclones (Flaounas et al., 2022) or upper-level cut-off lows in the western Mediterranean (Portmann et al., 2021)."*

    2.    Page 7, line 29: Together with the above major comments, please explain why you use 1991-2020 as reference period for the circulation anomalies and not the same as dust transport. Did you consider that maybe there are differences in atmospheric circulation due to interdecadal variability?

In climatological studies, climatological normals (ie. a 30-year period often the one encompassing the last three complete decades) is used as a reference period. Climatological Normals have long filled two major purposes. Firstly, they form a benchmark or reference against which conditions (especially current or recent conditions) can be assessed, and secondly, they are widely used (implicitly or explicitly) as an indicator of the conditions likely to be experienced

in a given location. This is also recommended by the World Meteorological Organization (WMO, see https://community.wmo.int/en/wmo-climatological-normals, see WMO 2007; https://library.wmo.int/records/item/52499-the-role-of-climatological-normals-in-a-changing-climate?offset=45), and this is what we have done to obtain the anomalies of the meteorological fields for the normal (2003-2019) and the anomalous (2020-2022) dust period.

We have confirmed that the results of the atmospheric circulation analyses in Section 3.2 are robust to the definition of the reference period (also if we use the 2003-2022 dust period as baseline for the atmospheric analyses). However, we have maintained the 1991-2020 period, as recommended by the WMO. This has been stated in the revised manuscript as follows:

*"Following international benchmark standards to assess changes in the meteorological conditions (WMO, 2007), we use 1991-2020 as the climatological normal period of analysis and the baseline to compute the anomalies of the meteorological fields. Although this reference period is not entirely covered by the MODIS observations employed for the characterisation of dust events (Section 2.1), the results of the atmospheric circulation analyses are robust to the study period (e.g. the 2003-2022 dust period) and the atmospheric reanalysis employed (e.g. ERA5; Hersbach et al., 2020)"*.

3. Page 10, Figure 1: It would be interesting to see how dust transport (and thus dust events) are distributed in other months as well, at least during December and April; the former to complete the "winter" and the latter to see is going on during the most active month regarding Mediterranean dust events.

Following the Referee's suggestion, Figure R1.3 shows the monthly mean AOD series for the period 2003-2022, including the months of extended winter seasons (December-to-March) and April. As expected, the mean AOD values and their interannual variations are higher in April than in the extended winter months, since dust intrusions become frequent in western Mediterranean by spring (i.e. Gkikkas et al., 2013). Despite this, AOD values were substantially larger in March than in April 2022 and slightly higher in 2020 and 2021, confirming the anomalous character of late winter 2020-2022. Figure R1.3 also indicates that the large AOD deviations of March 2020-2022 did not extend towards the spring, supporting our choice. This has been stressed in Section 2.1 of the revised manuscript as follows:

*"Although the climatological mean AOD over western Mediterranean increases in spring, the AOD values in March 2020-2022 were larger than in April 2020-2022, indicating that the anomalous character of the late winter 2020-2022 did not extend towards the spring"*.

Accordingly, the month of April has not been included in Figure 1 of the revised manuscript to keep the focus of the study on wintertime dust intrusions in the western Mediterranean. See also Major Comment #3.

[Figure]

*Figure R1.3. As Figure 1 (and Figure R1.1) of the main manuscript but including April.*

4. In addition, from this Figure, we can see that January does not exhibit any exceptional behaviour, while the period 2003-2005 was also quite active for February and March. Did you check what happened regarding circulation during those years? And seizing upon this, I would like to see if there were other "active" years for the entire period of MERRA-2. I am not convinced that 2020-2022 was indeed extreme.

Following the Referee's suggestion, we have analysed the February-March time series of MERRA-2 Dust AOD at 550nm of since 1980 (Figure R1.4). Although the consistency of this long dust-AOD-related series cannot be assured (see Major Comment #2), it would suggest that there has not been a dust-anomalous period such as that recorded in 2020-2022 since at least 1980. This has been stated in Section 3.1 of the revised manuscript as follows:

*"Additional analyses based on MERRA-2 (see section S2.1 of the Supplement) confirm that FM 2020-2022 over WEM was unprecedented since at least 1980 "*.

Of course, there were other multi-year periods with high AOD values in the past, however 2020-2022 shows an increase trend during the 4-year period including the maximum AOD of the entire time series. Note also that high monthly mean AOD values do not guarantee an anomalous frequency of dust intrusions. For example, the 2003-2005 February-March period mentioned by the reviewer only recorded a single dust intrusion in February.

[Figure]

*Figure R1.4. (Figure S2.4 of the Supplement) Time series of averaged dust-AOD at 550nm from MERRA-2 for February-March (FM) of 1980-2023 over the region [35°-50°N, 20°W-5°E].*

On the other hand, the boxplots of Figure R1.5 show similar (only slightly lower) AOD values for 1980-2002 period than for the 2003-2019 period, which supports the choice of the latter as a normal dust period in our study. Figure R1.5 also displays a pronounced increase in AOD between 2020 and 2022, being significantly different from the normal-dust period, and any other reference period considered. These results stress again the degree of exceptionality of February-March 2020-2022, supporting our definition of anomalous dust period.

[Figure]

*Figure R1.5. Dust-AOD boxplots from MERRA-2 for the region [35°-50°N, 20°W-5°E] and the February-March (FM) months considering five periods: 1980-2002, 2003-2019, 1980-2019, 1980-2022 and 2020-2022. Lower and upper boundaries for each box are the 25th and 75th percentiles; the red line is the median value; the dot blue is the mean value; and hyphens are the maximum and minimum values.*

5. Page 12, line 14: Is this correlation statistically significant?

All correlations are statistically significant (see Table R1). We have included this information in the Table S4 of the Supplement.

6. Page 14, Figure 3: Again, January has just 1 event in the "extreme period" and only 2 in the whole period. March exhibits a clear extreme behaviour during the "extreme period," and February to a lesser extent. I would suggest investigating the behaviour of dust transport in all months and I would strongly suggest investigating if there were such events over the whole 40 years period of MERRA-2. Then, you may focus on February and March only.

As explained before (see Major Comment #3), in the new version all analyses after Figure 2 have been limited to February and March. Accordingly, dust intrusions of December and January are no longer considered in the revised manuscript. Please see also our replies to Specific Comments #3 and #4.

7. Page 15, table 2: the comparison between January 2003-2019 and January 2020-2022 is not fair, because the latter consists only of 1 event lasting 3 days, thus, giving 1 dust day per year compared to the 0.18 of the entire period.

The month of January has been removed in the revised manuscript. Please, see our reply to the comment #7.

8. Page 16, table 3: the two parts of the table compare an 18-year period with a 3-year period. Which is the statistical significance of the differences between them? Are they significant and where?

The difference of the AOD distributions and that of the anomalous 2020-2022 period is highly significant (p-value << 0.01, as inferred from a Wilcoxon rank sum test), supporting the choice of the anomalous period, as shown in Figure R1.6 (new Figure S3.1 of the Supplement).

[Figure]

*Figure R1.6: Boxplots of daily MODIS/Aqua AOD for three periods for the region [35°-50°N, 20°W-5°E]: All (2003-2022), normal period (2003-2019) and anomalous period (2020-2022). Lower and upper boundaries for each box are the 25th and 75th percentiles, the red line is the median value, the dot blue is the mean values, and hyphens are the maximum and minimum values. Results are based on AOD retrievals from MODIS.*

9. Page 17, Figure 4: If I understand correctly, the second line of figures gives the clustering results only for the dust days of the 2020-2022 year period. Is this number of days adequate to provide a robust clustering, mainly for the clusters 3 and 4? My impression is that the first two clusters are quite similar for both periods implying that these two weather regimes are responsible for the dust events, while the other two are spurious.

We do not consider that clusters #3 and 4 are spurious. We find a good correspondence with the different types of high-pressure systems identified in section 3.2.2.2. However, we agree that the number of days included in these two clusters is low, likely emphasising specific features of individual dust events, which would result in unstable patterns for the less populated (high order) clusters. Therefore, we have repeated the clustering analysis of February-March dust days by retaining only the first two clusters, as requested by the Reviewer. We have modified Figure 4 and the associated change accordingly. Note that the main conclusion of the manuscript remains.

10. Page 19, line 25: The absolute anomalies of Z200 are of course larger than Z500, as Z200 gets larger and more variable values. Except you want to stress out something else, so please clarify.

We agree with the referee that this sentence can be ambiguous. Following her/his suggestion, the paragraph in Section 3.2 has been revised as follows:

*"For all dust events, the vertical cross-section shows negative Z anomalies between 925 and 200 hPa (not shown). These Z anomalies are in all cases prominent at 200 hPa and weaken towards the surface, consistent with the typical signatures of upper-level cut-off lows (Nieto et al., 2005, and references herein). Indeed, more than half of the FM dust events of the normal-dust 2003-2019 period concurred with a cut-off low".*

11. Page 20, line 20: How do we know that they are "intense enough to generate wind speed exceeding the threshold"?

Thank you very much for this question. In the submitted version we omitted to mention that for each dust event analysed in this study, the dust hotspots could be determined with a high spatial (3km x 3km) and temporal (15 min) resolution from EUMETSAT RGB dust product (Met Office; EUMETSAT, 2022; https://navigator.eumetsat.int/product/EO:EUM:DAT:MSG:DUST).

At least for the three case studies, we have first identified the dust hotspot using the RGB dust animations (see associates Timelapses animations at https://repositorio.aemet.es/handle/20.500.11765/15054), and then double-checked the surface wind speeds from the 6h NCEP/NCAR reanalysis at 0, 6, 12 and 18 UTC are within the windspeed threshold range to activate dust sources given by Helgren and Prospero (1987). However, a precise determination of geographic location as well as activation/deactivation time of each dust hotspot for each dust event analysed in this study would be an enormously tedious task and its results are outside the scope of this study.

We have included the following information at the end of Section 2.1:

*"The identification of dust hotspots was performed using the EUMETSAT RGB dust product (Met Office; EUMETSAT, 2022). This dataset contains RGB dust images from Meteosat Second Generation satellites over the full disc at a frequency of 15 minutes",*And in section 3.2 of the revised manuscript, we have added the following text:

*"For the three case studies analysed, the geographical location as well as the activation time of each dust hotspot was identified manually (Schepanski et al. 2007, 2009, 2012) by using the 15-*

*min EUMETSAT RGB dust animations (see Timelapse#1, 2, and 3 at https://repositorio.aemet.es/handle/20.500.11765/15054)."*

On the other hand, the range of threshold wind speeds to activate dust sources in Western Sahara given by Helgren and Prospero (1987), and still valid for modellers, should be taken as an approach since it was obtained as an average of measurements carried out during a month and a half (specifically from July 1 to August 15, 1974) at 12 UTC in eight stations located in Western Sahara, therefore they might not be applicable to arid Nort Africa in all seasons and years as the authors admit. Therefore, if in much of our dust hotspots geographical domain, 6-hourly NCEP/NCAR reanalysis surface wind speeds are within the range of threshold wind speeds for each day of each dust event should be sufficient to prove consistency between the experimental data on wind velocity threshold to activate dust hotspots and the real ability of cut-off lows to mobilize dust on ground. We have included in the revised manuscript the following text in Section S1 of the Supplement:

*"This is supported by 6-hourly surface wind data from NCEP/NCAR reanalysis over Morocco, Algeria, Western Sahara and Mauritania (see Section S1 of the Supplement)".*

12. Page 20, lines 19-20: Isn't it expected since January has only 1 event?

January is no longer analysed in the revised manuscript. Please, see our reply to Major Comment #1 and Specific Comments #6 and #7.

13. Page 20, line 28: Where do we see that probability of blocking doubles?

This is inferred from Figure 6a, which shows the climatological (in contours) and dust-conditioned (shading) blocking frequency, and the explanations given in page 20, lines 23-29. For February-March, the climatological frequency of blocking is ~6% (contours). Differently, during dust days of February-March, blocking frequency increases to >12% (red colour shading). Therefore, the probability of blocking occurrence at least doubles during dust days as compared to the climatology. We have clarified this in Section 3.2 of the revised manuscript:

*"Therein, the probability of blocking occurrence increases up to ~12% (red shading in Figure 6a), which more than doubles the expected values from the climatology (contours in Figure 6a)".*

14. Page 21, Figure 6 and respective discussion: From Figure 6b, it seems that not only the blocking activity is higher than climatology, is even lower. So, how does this affect the discussion about the extreme 2020-2022 years? I see from Figure 6d, a more zonal configuration of the jet. How could this help the northward dust transfer?

As indicated by the reviewer, blocking activity in February-March 2020-2022 was not significantly higher than the climatology. In some regions (e.g. western Mediterranean and southern Europe) blocking frequency was enhanced, whereas other regions reported a non-significant decrease. Therefore, the anomalous frequency of dust days in 2020-2022 cannot be fully explained by an unusual blocking activity. This was already stated in the original version of the manuscript (page 21, lines 13-15): *"blocking activity was not significantly higher than the climatology, suggesting that the anomalous frequency of dust days in 2020-2022 cannot be fully explained by a corresponding blocking increase over the favourable region for dust intrusions"*.

The lack of significant anomalies in blocking occurrence motivated the analysis of Figure 7, which aims to address if weather systems, other than blocking, contributed to the high frequency of dust intrusions in 2020-2022. Based on this analysis, we found a substantial number of dust days associated with high-pressure systems at low-latitudes, suggesting that subtropical ridges were relevant for the unusual activity of dust events in February-March 2020-2022. This was also stated in Section 3.2 the original version of the manuscript *"This non-blocking pattern was recurrent during 2020-2022 (~44% of the dust days) and it actually concurred with more dust days than in the historical period, explaining the reduced intervention of blocking"* and *"Therefore, while blocking was still the dominant pattern associated with dust days in 2020-2022, an enhanced occurrence of subtropical ridges and cut-off lows at low latitudes also contributed to the outstanding frequency of dust intrusions"*.

The more zonal configuration of the jet mentioned by the reviewer is congruent with these results, since subtropical ridges are associated with a reinforcement of the zonal wind in their poleward flanks and hence a more zonal and stronger eddy-driven jet in mid-latitudes (e.g. Woollings et al., 2010; Sousa et al., 2018; Barriopedro et al., 2023). The clustering analysis of dust days also confirms that dust intrusions can be associated with low pressure systems over the North Atlantic and hence zonal flow configurations (see e.g. the cluster #1 for 2003-2019 in Figure 4). Finally, the inspection of case studies also confirmed that dust transport can occur upstream of a subtropical ridge, presumably by the associated meridional flow, which can also be reinforced by an accompanying cut-off low. We have tried to clarify this further in Section 3.2 of the revised version of the manuscript:

*"These results indicate that dust events can be favoured by high-pressure systems at very different latitudes, ranging from subtropical ridges with mid-latitude jets to high-latitude blocks with poleward-shifted jets.  The occurrence of subtropical ridges was particularly enhanced during dust days of FM 2020-2022, weakening the strong block-dust linkage reported over the historical period".*

15. Page 21, lines 15-16: What do you mean by the phrase: "However, the spatial pattern hinders important intra-seasonal differences"? Why is this relevant here?

There are marked differences between February and March 2020-2022 (see Figure R1.7). In February 2020-2022, blocking activity was almost suppressed over central Europe, and increased over the Mediterranean, resulting in intensification of the jet at mid-latitudes. March 2020-2022 was characterized by poleward jets and enhanced blocking over the climatological region of occurrence. These intra-seasonal contrasts weaken the signal and significance shown in Figure 6 for February-March and motivate the description of monthly patterns in the text. In the revised text, we have rephrased the sentence, including this new Figure R1.7 in the Supplement to support our explanations:

*"However, the lack of statistical significance in the blocking and jet frequency of FM 2020-2022 partially results from contrasting signatures between the corresponding patterns for February and March 2020-2022 (Figure S17 of the Supplement)"*.

[Figure]

*Figure R1.7. As Figures 6b, of the manuscript but for the months of February and March.*

16. Page 25, lines 10-13: In conjunction with the above comment, I apologize if I'm mistaken, but I don't see an enhanced blocking and poleward jet configuration during 2020-2022, at least a more prominent one than for the entire period. So, what triggered

these extreme events? I think that it would be interesting to see the synoptic configuration as well.

See our replies to previous Specific Comments #13 and #14. A recurrent synoptic signature was a cut-off low, which was present in almost all dust events. The causes of the anomalous cut-off low activity and/or its enhanced efficiency to trigger dust events during 2020-2022 are unknown. At larger spatial scales, we diagnosed two main driving factors: subtropical ridges and high-latitude blocks, which presented above normal activity in February and March, respectively. This has been stressed further in Section 3.2 the revised version:

*"Likewise, the occurrence of cut-off lows was a common signature of dust intrusions during the 2020-2022 period. They were accompanied by either high-latitude blocks over Europe (C#1; Figure 4c) or high-pressure systems at lower latitudes (C#2; Figure 4d)". [...] "These results suggest that cut-off lows are actively involved in dust intrusions. At larger spatial scales, the enhanced dust activity of the recent 2020-2022 period could partially be explained by a high frequency of favourable configurations, including both recurrent (high-latitude blocking) and uncommon (e.g. subtropical high-pressure systems) dust-related patterns of the 2003-2019 period". [...] "Indeed, the high frequency of dust days associated with high-pressure systems at low latitudes (different to blocking) suggests that these systems were particularly relevant for the anomalous frequency of dust events in 2020-2022".*

There is a strong relationship between the occurrence of cut-off lows and blocking in the climatology (e.g. Nieto et al., 2007), and our analyses suggest a similar correspondence between cut-off lows and subtropical ridges, particularly during the anomalous period. The ultimate question is what caused the anomalous activity of cut-off lows and associated subtropical ridges (blocking) in February (March). This is out of the scope of the manuscript. In the revised text we have encouraged additional analyses to address this question:

*"Dedicated studies are required to address the causes of the anomalous activity of cut-off lows and associated subtropical ridges (blocking) in February (March) 2020-2022".*

Please, also note that the synoptic patterns requested by the reviewer are already provided in Figure 7, as well as in Figure 4 (clustering analysis of dust days).

17. Page 27, line 10: The period 1958-1998 is totally different from the period used in the present study, thus, any trends found in the frequency of cut-off lows back then are not necessarily continue to the examined period.

We agree with the reviewer. Note, however, that this sentence was included in the discussion about long-term changes to stress that, to the best of our knowledge, there are no robust

evidence of anthropogenic influences in cut-off lows. Of course, this lack of evidence should not be taken as evidence of no change (forced trends might have not emerged at that time, and one could find significant trends in a more recent period). In the revised version we have stressed that the results of Nieto et al. (2007) are not informative of our period of analysis, which calls for updated analyses of wintertime trends in cut-off lows over the European sector:

*"Nieto et al. (2007) did not report significant trends in the frequency of winter cut-of lows over the European sector for the 1958-1998 period. Although this period does not inform on the last decades (with stronger anthropogenic forcing) or our study period, it suggests weak cut-off low responses to long-term climate change. However, the key role of Euro-Atlantic cut-off lows in WEM dust dust activtycalls for updated analyses of their trends and variability".*

**Minor comments**

1.  Page 4, lines 12-13: A reference is needed here.

We have included Flaounas et al. (2015), as requested.

2.  Page 7, line 18: "as the anomaly of the projection". Please clarify which projection you refer to.

We mean the spatial projection of the instantaneous Z field onto the centroid of the respective cluster, Zc. Mathematically, this is computed as the dot product <Z,Zc>. This has been clarified in the revised manuscript, as described in our reply to the next comment.

3.  Page 7, line 19: "and all indices of that day". Then what? The phrase is incomplete.

We agree with the Referee. In order to complete this information, we have replaced the text in Section 3.1 by the following sentences:

*"These WRs are derived from a k-means clustering in the phase space spanned by the seven leading empirical orthogonal functions of 10-day low-pass filtered Z500\* fields, with Z500\* denoting the normalized 500 hPa geopotential height anomaly over the Euro-Atlantic sector. Following Michel and Rivière (2011), for each day and WR we compute a WR index as the spatial projection of the daily unfiltered Z500\* onto the cluster centroid (i.e. the mean Z500\* for all days in the cluster). The resulting indices are normalised (zero mean and one SD). A day is assigned to a given WR if the respective index is greater than 1 and higher than that of all other WRs".*

4.  Page 12, line 18: I would say it is 1.6 dust events/year.

The sentence has been rewritten in the revised manuscript as follows:

*"According to the AOD thresholds specified in Table 1, during FM 2003-2022 we have identified a total of 30 dust events (1.5 dust-events/year) over WEM".*

5.  Page 19, line 21: You mean composite anomalies?

This has been clarified in Section 3.2 of the revised manuscript. This sentence reads now as follows:

*"The analysis relies on composite anomalies for the 2003-2019 and 2020-2022 period, separately, to emphasise distinctive features of the recent anomalous period".*

6.  The literature cited in the text and in the references is poorly prepared with many mistakes and omissions making it very difficult for the reader to keep track. Please find below a non-exhaustive list:

We appreciate, and regret at the same time, the time spent by Reviewer #1 in identifying citation errors. In the revised version, we have carefully revised the bibliography.

7.  Page 2, line 28: Kuula et al. 2021 wrong year?

Corrected to Kuula et al. (2022)

8.  Page 4, Line 11: correct kikas to Gkikas

Amended.

9.  Page 6, Line 21: Liu et al. 2009 or 2019?

In this case both citations are correct. They have been rearranged in the bibliography section.

10. Page 12, line 11: Moulin et al. 1998 or 1997?

Moulin et al. (1998) is correct. The reference Moulin et al. (1997) has been removed in the revised manuscript.

11. Page 28, line 23; page 30, line 8; page 30, line 34; page31, line 14; page 32, line 30; page 33, line 14; page34, line 34; page 36, line 23; the paragraphs are merged.

Amended.

12. References Barnaba et al., Gelaro et al., Hersbach et al., Holben et al., Klose et al., Labban et al., Liu et al., Munoz et al., Ryder et al., Schepanski et al. are not cited in the manuscript.

After the revision, we have checked the list of references and taken the following actions:

a.  Barnaba et al. (2022) has been removed.

b. Gelaro et al. (2017) has been removed.

c. Holben et al. (1998) has been moved to supplementary material.

d. Klose et al. (2021) has been removed.

e. Liu et al. (2009) and Liu et al. (2019) are cited in the manuscript.

f. Muñoz et al. (2020) is cited in the manuscript.

g. Ryder at al. (2018) is cited in the manuscript.

h. Schepanski et al. (2007 and 2009) are cited in the manuscript.

**References**

*Cuevas, E., Gómez-Peláez, A. J., Rodríguez, S., Terradellas, E., Basart, S., García, R. D., García, O.E. & Alonso-Pérez, S. (2017). The pulsating nature of large-scale Saharan dust transport as a result of interplays between mid-latitude Rossby waves and the North African Dipole Intensity. Atmospheric environment, 167, 586-602.*

*Heidinger, A. K., Foster, M.J., Walther, A., and Zhao, X.: The Pathfinder Atmospheres–Extended AVHRR climate dataset. Bull. Amer. Meteor. Soc., 95, 909–922, doi:10.1175/BAMS-D-12-00246.,2014.*

*Holben, B., and Coauthors, 1998: AERONET—A federated instrument network and data archive for aerosol characterization. Remote Sens. Environ., 66, 1–16, doi:10.1016/S0034-4257(98)00031-5.*

*Met Office; EUMETSAT (2022): MSG: Dust imagery in the RGB channels over the full disc at 41.5 degrees East (LEDF41, upto 0900 UTC 1st June 2022). NERC EDS Centre for Environmental Data Analysis, date of citation. https://catalogue.ceda.ac.uk/uuid/b1dacc09b42f4d8ab492c5d5c751efa9.*

*Randles, C. A., da Silva, A. M., Buchard, V., Colarco, P. R., Darmenov, A., Govindaraju, R., Smirnov, A., Holben, B., Ferrare, R., Hair, J., Shinozuka, Y., and Flynn, C. J.: The MERRA-2 Aerosol Reanalysis, 1980 Onward. Part I: System Description and Data Assimilation Evaluation, Journal of Climate, American Meteorological Society, 30, 17, 6823-6850, 0894-8755, https://journals.ametsoc.org/view/journals/clim/30/17/jcli-d-16-0609.1.xml, https://doi.org/10.1175/JCLI-D-16-0609.1, 2017.*

*Real, E., Couvidat, F., Ung, A., Malherbe, L., Raux, B., Gressent, A., and Colette, A.: Historical reconstruction of background air pollution over France for 2000–2015, Earth Syst. Sci. Data, 14, 2419–2443, https://doi.org/10.5194/essd-14-2419-2022, 2022.*

*Schepanski, K., Tegen, I., Laurent, B., Heinold, B., and Macke,A.: A new Saharan dust source activation frequency map derived from MSG-SEVIRI IR-channels, Geophys. Res. Lett., 34, L18803, https://doi.org/10.1029/2007GL030168, 2007.*

*Schepanski, K., Tegen, I., and Macke, A.: Comparison of satellite based observations of Saharan dust source areas, Remote Sens. Environ., 123, 90–97, ISSN 0034-4257, https://doi.org/10.1016/j.rse.2012.03.019 , 2012.*

*Schmetz, J., Pili, P., Tjemkes, S., Just, D., Kerkmann, J., Rota, S., and Ratier, A: An introduction to Meteosat Second Generation (MSG), Bull. Am. Meteorol. Soc., 83, 977– 992, https://doi.org/10.1175/1520-0477(2002)083<0977 :AITMSG>2.3.CO;2, 2002.*